# Wavelet Canonical Coherence for Nonstationary Signals

**Haibo Wu**[1]*  **Marina I. Knight**[2]
**Keiland W. Cooper**[3,4]   **Norbert J. Fortin**[3,4]   **Hernando Ombao**[1]

[1]Statistics Program, King Abdullah University of Science and Technology
[2]Department of Mathematics, The University of York
[3]Department of Neurobiology and Behavior, University of California, Irvine
[4]Center for the Neurobiology of Learning and Memory, University of California, Irvine

## Abstract

Understanding the evolving dependence between two sets of multivariate signals is fundamental in neuroscience and other domains where sub-networks in a system interact dynamically over time. Despite the growing interest in multivariate time series analysis, existing methods for between-clusters dependence typically rely on the assumption of stationarity and lack the temporal resolution to capture transient, frequency-specific interactions. To overcome this limitation, we propose scale-specific wavelet canonical coherence (WaveCanCoh), a novel framework that extends canonical coherence analysis to the nonstationary setting by leveraging the multivariate locally stationary wavelet model. The proposed WaveCanCoh enables the estimation of time-varying canonical coherence between clusters, providing interpretable insight into scale-specific time-varying interactions between clusters. Through extensive simulation studies, we demonstrate that WaveCanCoh accurately recovers true coherence structures under both locally stationary and general nonstationary conditions. Application to local field potential (LFP) activity data recorded from the hippocampus reveals distinct dynamic coherence patterns between correct and incorrect memory-guided decisions, illustrating the capacity of the method to detect behaviorally relevant neural coordination. These results highlight WaveCanCoh as a flexible and principled tool for modeling complex cross-group dependencies in nonstationary multivariate systems. Code for implementing WaveCanCoh is available at https://github.com/mhaibo/WaveCanCoh.git.

## 1   Introduction

Assessing the dependence structure between node clusters in a network is one of the most critical aspects of network time series analysis. Many models and frameworks have been developed to capture between-clusters association (e.g., correlation, coherence, and causality). Most existing methods characterize the dependence between two clusters through the dependence between (many) node pairs. However, in many scenarios, the primary interest lies in understanding the dependence structure between two groups of multivariate time series rather than individual processes. Figure 1 illustrates this perspective using brain activity signals. In this experiment, local field potential (LFP) activity was recorded from multiple electrodes implanted in different subregions of the hippocampus of rodents (rats) as they performed a complex sequence memory task. Instead of focusing on coherence between individual channels (electrodes), the main goal is to quantify time-varying functional interactions between groups of electrodes in order to understand how information processing differs in these two

---

*Corresponding author: `haibo.wu@kaust.edu.sa`

subregions. Similar challenges arise in other domains. For instance, in finance, understanding the dependence between entire market sectors (e.g., technology and energy) can be more informative than analyzing associations between individual stocks. These scenarios require a framework capable of capturing dynamic coherence between sets of nonstationary multivariate signals. In this paper, we propose a novel framework called scale-specific wavelet canonical coherence (WaveCanCoh) to characterize time-localized and scale-specific coherence between two clusters of multivariate time series. By leveraging the time-frequency localization properties of wavelets, WaveCanCoh is well-suited for analyzing nonstationary multivariate signals in neuroscience, finance, and other fields where transient, cross-group interactions are of scientific interest.

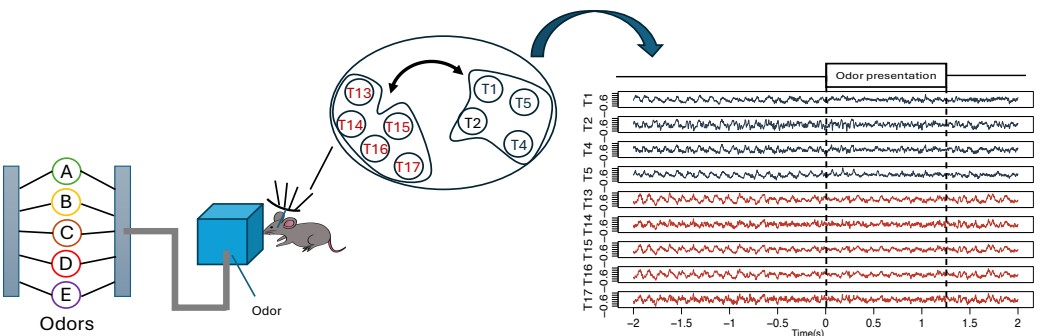

Figure 1: Schematic overview of the experimental and analytical motivation. Local field potential (LFP) activity was recorded from multiple electrodes in the hippocampus as rats performed an odor sequence memory task. Rather than focusing on individual electrode pairs, our goal is to characterize the dynamic dependence between two hippocampal subregions, each represented by a group of electrodes, using wavelet-based multivariate coherence.

Canonical variate analysis (CVA) ([14]) provides a method for measuring the correlation between two vector variables, its application to time series data having started in the 1950s and 1960s, primarily in the econometrics and signal processing fields. [3] and [10] extend CVA to time series for forecasting and causality detection. [4] provides a spectral domain formulation of canonical correlation, useful for frequency-domain time series analysis. This approach enables the analysis of canonical coherence between two sets of time series across different frequency bands. Many modern studies have been developed within this framework (e.g., [24] and [25]), and related methods have been widely applied across various fields, including neuroscience, finance, speech processing, and machine learning.

However, previous methods rely on the assumption that time series are (weakly) stationary, meaning their statistical properties (e.g., expectation and covariance, spectral signature) remain constant over time. In practice, this assumption often does not hold for time series that arise in practice. Moreover, two sets of time series commonly exhibit time-varying global coherence, which can sometimes be crucial for analysis. Thus, a method capable of handling nonstationary time series is necessary. Wavelet analysis is a widely used tool for studying nonstationary time series, as its localization property allows for the examination of localized correlations between two time series across both time and frequency domains. Wavelets are particularly effective for capturing transient properties of nonstationary signals [13] due to their compact support, which can be compressed or stretched to adapt to the dynamic characteristics of the signal. Wavelet coherence has been well-defined and extensively studied in previous research, with applications spanning various fields [9, 12].However, prior studies primarily focus on within-network coherence in multivariate systems or pairwise coherence between univariate channels [5, 6], and several wavelet-based connectivity approaches have been proposed for fMRI [23] and MEG [11]. To the best of our knowledge, no existing work has extended classical canonical coherence to the wavelet domain to measure the time-evolving canonical coherence between two groups of multivariate time series.

The key novelty of this paper lies in the development of a comprehensive and rigorous framework based on wavelets for measuring canonical coherence between two sets of nonstationary multivariate time series. Specifically, our main contributions include: (1) we define scale-specific wavelet canonical coherence (WaveCanCoh) and introduce its use as a tool to quantify the coherence between two sets of multivariate time series, (2) we provide a complete and theoretically justified algorithm for its estimation, and (3) we apply our method to LFP activity data from multiple electrodes to

quantify dynamic interaction patterns among different subregions in the hippocampus. Multivariate locally stationary wavelet processes (MvLSW) underpin our construction, and the reader is directed to [19] and [21] for details on their construction. Our framework not only captures the time-varying coherence between two sets of signals but also determines the contribution of each individual channel to the global coherence. Compared to previous models, the proposed approach provides a detailed, localized characterization of interactions within the multivariate time series. Our findings on the LFP activity data offer new insights into the functional relationship between hippocampal subregions, demonstrating the potential of our method to advance the study of functional brain connectivity.

The format of the paper is as follows. Section 2 overviews the current methodology for assessing time series canonical coherence. Section 3 provides a brief overview of MvLSW processes, supporting a detailed introduction to our proposed WaveCanCoh framework. Its estimation, and that of related parameters, is tackled in Section 4. Section 5 validates the proposed framework and demonstrates its performance through simulation. In Section 6 we apply the WaveCanCoh method on LFP activity data collected from rats to investigate the dynamic interactions of different subregions in the hippocampus during memory tasks. Section 7 concludes the paper, with the Appendix offering further theoretical and empirical supporting information, as well as reflections on the method's limitations in Appendix F.

## 2   Related works

First, we provide a brief introduction to classical canonical correlation analysis for time series, with the primary goal to characterize dependence between two clusters of time series, where each cluster features several nodes. In particular, we consider two multivariate time series with dimensions $p$ and $q$ respectively, denoted by $\mathbf{X}_t = \left( X_t^{(1)}, \ldots, X_t^{(p)} \right)^\top$ and $\mathbf{Y}_t = \left( Y_t^{(1)}, \ldots, Y_t^{(q)} \right)^\top$, for $t = \{1, \ldots, T\}$. Typically, $\{\mathbf{X}_t\}$ and $\{\mathbf{Y}_t\}$ are assumed to be zero-mean weakly stationary time series. Letting $\mathbf{Z}_t$ denote the concatenated $(p + q)$ dimension time series, $\mathbf{Z}_t = \left( X_t^{(1)}, \ldots, X_t^{(p)}, Y_t^{(1)}, \ldots, Y_t^{(q)} \right)^\top$, its covariance matrix at lag $\tau$ is

$$\mathbf{\Sigma_{ZZ}}(\tau) = \left( \begin{array}{cc} \mathbf{\Sigma_{XX}}(\tau) & \mathbf{\Sigma_{XY}}(\tau) \\ \mathbf{\Sigma_{YX}}(\tau) & \mathbf{\Sigma_{YY}}(\tau) \end{array} \right),$$

where $\mathbf{\Sigma_{XX}}(\cdot)$, $\mathbf{\Sigma_{YY}}(\cdot)$ are the autocovariance matrices of $\{\mathbf{X}_t\}$ and $\{\mathbf{Y}_t\}$ respectively, and $\mathbf{\Sigma_{XY}}(\cdot)$, $\mathbf{\Sigma_{YX}}(\cdot)$ are their cross-covariances. The canonical correlation between $\{\mathbf{X}_t\}$ and $\{\mathbf{Y}_t\}$ at lag $\tau$, $\boldsymbol{\rho}(\tau)$, is defined as

$$\boldsymbol{\rho}(\tau) = \max_{\mathbf{a}, \mathbf{b}} \frac{\mathbf{a}^\top \mathbf{\Sigma_{XY}}(\tau) \mathbf{b}}{\sqrt{\mathbf{a}^\top \mathbf{\Sigma_{XX}}(\tau) \mathbf{a}} \sqrt{\mathbf{b}^\top \mathbf{\Sigma_{YY}}(\tau) \mathbf{b}}}, \tag{1}$$

where $\mathbf{a} \in \mathbb{R}^p$ and $\mathbf{b} \in \mathbb{R}^q$ are called canonical correlation vectors, subject to standardized constraints $\mathbf{a}^\top \mathbf{\Sigma_{XX}} \mathbf{a} = 1$, $\mathbf{b}^\top \mathbf{\Sigma_{YY}} \mathbf{b} = 1$ [4]. Thus, the canonical correlation can be rewritten as

$$\boldsymbol{\rho}(\tau) = \max_{\mathbf{a}, \mathbf{b}} \left( \mathbf{a}^\top \mathbf{\Sigma_{XY}}(\tau) \mathbf{b} \right). \tag{2}$$

***Remark 1****: In the preceding definitions for the cross-covariance matrices, we have $\mathbf{\Sigma_{XY}}(\tau) = \mathbf{\Sigma_{YX}}(-\tau) = \mathbb{E}\left[ \mathbf{X}_t \mathbf{Y}_{t-\tau}^\top \right]$, which measure the lagged covariance between $\{\mathbf{X}_t\}$ and $\{\mathbf{Y}_t\}$, i.e., past values of $\mathbf{Y}$ may be associated to present values of $\mathbf{X}$ for $\tau > 0$, and vice-versa when $\tau < 0$. The case of $\tau = 0$ illustrates contemporaneous relationships.*

The solution for $\mathbf{a}$ and $\mathbf{b}$ in equation (1) can be obtained from the eigenvectors of the following matrices, respectively (in what follows, $\tau$ is dropped for brevity),

$$\mathbf{\Sigma_{XX}^{-1}} \mathbf{\Sigma_{XY}} \mathbf{\Sigma_{YY}^{-1}} \mathbf{\Sigma_{YX}}, \text{ and} \tag{3}$$

$$\mathbf{\Sigma_{YY}^{-1}} \mathbf{\Sigma_{YX}} \mathbf{\Sigma_{XX}^{-1}} \mathbf{\Sigma_{XY}}. \tag{4}$$

Here, $\mathbf{a}$ and $\mathbf{b}$ are the eigenvectors corresponding to the largest eigenvalue, $\lambda$, of matrices (3) and (4), respectively, and for largest canonical correlation coefficient we have $\boldsymbol{\rho} = \sqrt{\lambda}$ [17]. The framework above yields the canonical correlation between $\{\mathbf{X}_t\}$ and $\{\mathbf{Y}_t\}$. However, in many practical cases, such as EEG analysis, canonical correlation in the spectral domain is more meaningful than in the

time domain, as components at different frequencies (or scales) reveal crucial information about neural dynamics and functional connectivity [20] , [4] extends the time-domain canonical correlation into the spectral domain). Namely, suppose the spectral matrix of $\mathbf{Z}_t = \left(\mathbf{X}_t^\top, \mathbf{Y}_t^\top\right)^\top$ is

$$\mathbf{f_{ZZ}}(\boldsymbol{\omega}) = \left[\begin{array}{cc} \mathbf{f_{XX}}(\boldsymbol{\omega}) & \mathbf{f_{XY}}(\boldsymbol{\omega}) \\ \mathbf{f_{YX}}(\boldsymbol{\omega}) & \mathbf{f_{YY}}(\boldsymbol{\omega}) \end{array}\right],$$

where $\mathbf{f_{XX}}(\boldsymbol{\omega})$ is the $p \times p$ autospectral matrix of $\{\mathbf{X}_t\}$, $\mathbf{f_{YY}}(\boldsymbol{\omega})$ is the $q \times q$ autospectral matrix of $\{\mathbf{Y}_t\}$, and $\mathbf{f_{XY}}(\boldsymbol{\omega})$ is the $p \times q$ cross-spectral matrix between $\{\mathbf{X}_t\}$ and $\{\mathbf{Y}_t\}$. Given vectors $\mathbf{a} \in \mathbb{C}^p$ and $\mathbf{b} \in \mathbb{C}^q$, such that $\mathbf{a}^\top \mathbf{f_{XX}}(\boldsymbol{\omega})\mathbf{a} = \mathbf{b}^\top \mathbf{f_{YY}}(\boldsymbol{\omega})\mathbf{b} = 1$, the canonical coherence at frequency $\boldsymbol{\omega}$ is

$$\boldsymbol{\rho}(\boldsymbol{\omega}) = \max_{\mathbf{a},\mathbf{b}} \left| \frac{\mathbf{a}^\top \mathbf{f_{XY}}(\boldsymbol{\omega})\mathbf{b}}{\sqrt{\mathbf{a}^\top \mathbf{f_{XX}}(\boldsymbol{\omega})\mathbf{a}}\sqrt{\mathbf{b}^\top \mathbf{f_{YY}}(\boldsymbol{\omega})\mathbf{b}}} \right|^2. \tag{5}$$

By solving the maximization problem in equation (5), the canonical coherence vectors $\mathbf{a}$ and $\mathbf{b}$ are determined, leading to the quantification of the canonical coherence at frequency $\boldsymbol{\omega}$.

Note the classic canonical coherence in (5) completely ignores temporal dynamics, a consequence of the stationarity assumption where dependence between clusters is imposed to remain constant over time, whilst most real-world data, such as EEG, exhibit nonstationarity [15, 16, 26]. Hence the lack of time-localization information in the above approach may result in misleading results and a novel method capable of capturing time-varying canonical coherence is needed.

## 3   Wavelet canonical coherence (WaveCanCoh)

Our WaveCanCoh framework is built upon the multivariate locally stationary wavelet (MvLSW) process ([19], [21]), which is a model based on wavelet analysis for time series. A brief overview of wavelets and highlight of their differences from Fourier-based methods are provided in Appendix A.

A new representation for discretely sampled nonstationary time series based on discrete non-decimated wavelets is the locally stationary wavelet process introduced by [19], later extended to a multivariate framework in [21]. A $P$-variate stochastic process with time evolving second-order structure, $\mathbf{X}_t = \left(X_t^{(1)}, X_t^{(2)}, \ldots, X_t^{(P)}\right)^\top$, where $t = 1, \ldots, T$, can be represented with the MvLSW formulation

$$\mathbf{X}_t = \sum_{j=1}^\infty \sum_{k \in \mathbb{Z}} \mathbf{V}_j(k/T)\psi_{j,k}(t)\mathbf{z}_{j,k},$$

where $\mathbf{V}_j(k/T)$ is a $P \times P$ transfer function matrix assumed to have a lower-triangular form; $\{\psi_{j,k}\}_{j,k}$ is a set of discrete non-decimated wavelets; $\{\mathbf{z}_{j,k}\}_{j,k}$ is a set of $P \times 1$ uncorrelated random vectors with (column) mean vector $\mathbf{0}$ and $P \times P$ identity covariance matrix. Since the wavelet basis $\psi_{j,k}(t)$ is localized in both time and frequency, the transfer matrix $\mathbf{V}_j(k/T)$ provides a measure of the time-varying contribution to the variance among channels at a specific scale $j$ and rescaled time $u = k/T$, thus enabling the statistical properties of the process $\{\mathbf{X}_t\}$ to change smoothly over time.

The time-varying statistical properties of $\{\mathbf{X}_t\}$ can be captured through the localized, scale-specific local wavelet spectral matrix (LWS, [21]), $\mathbf{S}_j(u)$, defined at scale $j$ and rescaled time $u \in (0, 1)$, as

$$\mathbf{S}_j(u) = \mathbf{V}_j(u)\mathbf{V}_j^\top(u). \tag{6}$$

Note $\mathbf{S}_j(u)$ is a $P \times P$ symmetric, positive semi-definite matrix and its $(p, q)$ entry, $S_j^{(p,q)}(u)$, denotes the cross-spectrum between channels $p$ and $q$. We now extend the LWS matrix construction from a single set of multivariate time series to a cross-group LWS matrix, between $\mathbf{X}_t = \left(X_t^{(1)}, \ldots, X_t^{(P)}\right)^\top$ and $\mathbf{Y}_t = \left(Y_t^{(1)}, \ldots, Y_t^{(Q)}\right)^\top$. Denoting $\mathbf{Z}_t = \left(\mathbf{X}_t^\top, \mathbf{Y}_t^\top\right)^\top$, the LWS matrix of $\{\mathbf{Z}_t\}$ at scale $j$ and rescaled time $u$, $\mathbf{S}_{j;\mathbf{ZZ}}(u)$, is

$$\mathbf{S}_{j;\mathbf{ZZ}}(u) = \mathbf{V}_{j;\mathbf{Z}}(u)\mathbf{V}_{j;\mathbf{Z}}^\top(u) = \left[\begin{array}{cc} \mathbf{S}_{j;\mathbf{XX}}(u) & \mathbf{S}_{j;\mathbf{XY}}(u) \\ \mathbf{S}_{j;\mathbf{YX}}(u) & \mathbf{S}_{j;\mathbf{YY}}(u) \end{array}\right]. \tag{7}$$

In equation (7), $\mathbf{V}_{j;\mathbf{Z}}(u)$ denotes the $(P+Q) \times (P+Q)$ transfer function matrix of the MvLSW process $\{\mathbf{Z}_t\}$, and $\mathbf{S}_{j;\mathbf{ZZ}}(u)$ is its corresponding LWS matrix. The main diagonal blocks $\mathbf{S}_{j;\mathbf{XX}}(u)$ $(P \times P)$ and $\mathbf{S}_{j;\mathbf{YY}}(u)$ $(Q \times Q)$ denote the auto-LWS matrices of the $\{\mathbf{X}_t\}$ and $\{\mathbf{Y}_t\}$ processes, respectively, while $\mathbf{S}_{j;\mathbf{XY}}(u)$ and $\mathbf{S}_{j;\mathbf{YX}}(u)$ denote their cross-LWS matrices.

***Remark 2:*** *In equation (7), $\mathbf{S}_{j;\mathbf{XY}}(u)$ is a $P \times Q$ matrix at each time point, and the $(p,q)$ element gives the cross-spectrum between channel $p$ of $\{\mathbf{X}_t\}$ and channel $q$ of $\{\mathbf{Y}_t\}$. Moreover, it is easy to show that $\mathbf{S}_{j;\mathbf{XY}}(u) = \mathbf{S}_{j;\mathbf{YX}}^\top(u)$.*

The LSW matrix quantifies the localized contributions to the process variance for individual and cross-channels, which motivates us to next define the localized canonical coherence between two sets of locally stationary time series at a specific scale (corresponding to a determined frequency band).

### Definition 1 (Localized Scale-specific Wavelet Canonical Coherence)

Let $\mathbf{X}_t = \left( X_t^{(1)}, \ldots, X_t^{(P)} \right)^\top$ and $\mathbf{Y}_t = \left( Y_t^{(1)}, \ldots, Y_t^{(Q)} \right)^\top$, where $t = 1, \ldots, T$, be (jointly) multivariate locally stationary time series. We define the localized scale-specific wavelet canonical coherence (WaveCanCoh) between $\{\mathbf{X}_t\}$ and $\{\mathbf{Y}_t\}$, at scale $j$ and rescaled time $u$, as

$$\boldsymbol{\rho}_{j;\mathbf{XY}}(u) = \max_{\mathbf{a}_j(u), \mathbf{b}_j(u)} \left\{ \mathbf{a}_j^\top(u) \mathbf{S}_{j;\mathbf{XY}}(u) \mathbf{b}_j(u) \right\}^2, \tag{8}$$

where $\mathbf{a}_j^\top(u) = \left( a_j^{(p)}(u) \right)_{p=1}^P$ is a $1 \times P$ vector and $\mathbf{b}_j^\top(u) = \left( b_j^{(q)}(u) \right)_{q=1}^Q$ is a $1 \times Q$ vector, representing the localized canonical coherence vectors of $\{\mathbf{X}_t\}$ and $\{\mathbf{Y}_t\}$, respectively. The constraints here are $\mathbf{a}_j^\top(u) \mathbf{S}_{j;\mathbf{XX}}(u) \mathbf{a}_j(u) = 1$ and $\mathbf{b}_j^\top(u) \mathbf{S}_{j;\mathbf{YY}}(u) \mathbf{b}_j(u) = 1$.

***Remark 3:*** *The WaveCanCoh time-dependent trace $\boldsymbol{\rho}_{j;\mathbf{XY}}(\cdot)$ measures the 'global' coherence between $\{\mathbf{X}_t\}$ and $\{\mathbf{Y}_t\}$ at scale $j$, and takes values between 0 and 1. A value close to 1 indicates strong linear dependence, while a value close to 0 shows little to no linear dependence. Furthermore, $\mathbf{a}_j^{(p)}(\cdot)$ and $\mathbf{b}_j^{(q)}(\cdot)$ represent the localized contributions from the $(p,q)$ channels to $\boldsymbol{\rho}_{j;\mathbf{XY}}(\cdot)$.*

The canonical coherence vectors $\mathbf{a}_j(\cdot), \mathbf{b}_j(\cdot)$ can be obtained by maximizing (8) and the solution can be found by solving the eigenvalue and eigenvector problem associated with the following matrices

$$\mathbf{M}_{j;\mathbf{a}}(u) = \mathbf{S}_{j,\mathbf{XX}}^{-1}(u) \mathbf{S}_{j,\mathbf{XY}}(u) \mathbf{S}_{j,\mathbf{YY}}^{-1}(u) \mathbf{S}_{j,\mathbf{YX}}(u), \tag{9}$$

$$\mathbf{M}_{j;\mathbf{b}}(u) = \mathbf{S}_{j,\mathbf{YY}}^{-1}(u) \mathbf{S}_{j,\mathbf{YX}}(u) \mathbf{S}_{j,\mathbf{XX}}^{-1}(u) \mathbf{S}_{j,\mathbf{XY}}(u). \tag{10}$$

Denote by $\Lambda_{j;\mathbf{a}}^{(k)}(u)$ the $k$-th largest eigenvalue of matrix $\mathbf{M}_{j;\mathbf{a}}(u)$ in equation (9), and by $\Lambda_{j;\mathbf{b}}^{(l)}(u)$ the $l$-th largest eigenvalue of matrix $\mathbf{M}_{j;\mathbf{b}}(u)$ in equation (10), for $k, l = 1, \ldots, \min(P, Q)$. An important observation is that $\mathbf{M}_{j;\mathbf{a}}(u)$ and $\mathbf{M}_{j;\mathbf{b}}(u)$ share the same eigenvalues (see Appendix B for details), hence denoting by $\Lambda_j^{(1)}(u) = \Lambda_{j;\mathbf{a}}^{(1)}(u) = \Lambda_{j;\mathbf{b}}^{(1)}(u)$ their largest eigenvalue, the canonical coherence between $\{\mathbf{X}_t\}$ and $\{\mathbf{Y}_t\}$ at rescaled time $u$, as defined in equation (8), becomes

$$\boldsymbol{\rho}_{j;\mathbf{XY}}(u) = \Lambda_j^{(1)}(u). \tag{11}$$

The eigenvectors of $\mathbf{M}_{j;\mathbf{a}}(u)$ and $\mathbf{M}_{j;\mathbf{b}}(u)$ corresponding to $\Lambda_j^{(1)}(u)$ provide the solutions to the canonical directions of $\{\mathbf{X}_t\}$ and $\{\mathbf{Y}_t\}$, respectively. *Proof: see Appendix B.*

An important extension of our framework is the incorporation of leading-lag relationships into the scale-specific wavelet canonical coherence (WaveCanCoh). To account for potential causal effects, we define a lagged joint process, $\mathbf{Z}_t(h) = \left( \mathbf{X}_t^\top, \mathbf{Y}_{t+h}^\top \right)^\top$, where $h$ is the value of lag, and $t = 1, \ldots, T-h$ for $h > 0$. We define the LWS matrix of $\{\mathbf{Z}_t(h)\}$ at scale $j$, as

$$\mathbf{S}_{j;\mathbf{ZZ}}(u,h) = \mathbf{V}_{j;\mathbf{Z}}(u,h) \mathbf{V}_{j;\mathbf{Z}}^\top(u,h) = \begin{bmatrix} \mathbf{S}_{j;\mathbf{XX}}(u) & \mathbf{S}_{j;\mathbf{XY}}(u,h) \\ \mathbf{S}_{j;\mathbf{YX}}(u+(h/T), -h) & \mathbf{S}_{j;\mathbf{YY}}(u+(h/T)) \end{bmatrix} \tag{12}$$

where $\mathbf{S}_{j;\mathbf{XY}}(u,h)$ denotes the cross-LWS matrix between $\mathbf{X}_{[uT]}$ and $\mathbf{Y}_{[uT]+h}$, capturing the interaction between current values of $\mathbf{X}$ and future values of $\mathbf{Y}$. Based on this construction, we can define and estimate the lagged version of WaveCanCoh, enabling us to infer potential causal relationships between two groups of time series.

**Definition 2 (Causal Localized Scale-specific Wavelet Canonical Coherence)**
The causal localized scale-specific canonical coherence (Causal-WaveCanCoh) between $\{\mathbf{X}_t\}$ and $\{\mathbf{Y}_t\}$ with lag $h$ (or, $\mathbf{X}_t \to \mathbf{Y}_{t+h}$), at scale $j$ and rescaled time $u$, is defined as

$$\boldsymbol{\rho}_{j;\mathbf{XY}}(u,h) = \max_{\mathbf{a}_j(u),\mathbf{b}_j(u)} \left\{ \mathbf{a}_j^\top(u)\mathbf{S}_{j;\mathbf{XY}}(u,h)\mathbf{b}_j(u+(h/T)) \right\}^2, \qquad (13)$$

where the notations and constraints are the same as in the standard WaveCanCoh framework. This extension allows for a scale- and time-specific evaluation of causal overall association between two sets of time series, enhancing interpretability in dynamic, multivariate, and nonstationary settings.

The framework above allows us to capture the time-varying overall association between two sets of multivariate time series, as well as the time-varying contributions from each individual channel within these sets. However, a natural consideration is how to project this time-varying coherence at each scale $j$ into the frequency domain in a manner consistent with the Fourier-based method described in Section 2, as a key concern in many analyses is to interpret the results in the frequency domain. As mentioned earlier, each scale in the wavelet analysis corresponds approximately, but not exactly, to a specific frequency band. This correspondence is governed by the unique filtering mechanism of wavelets, and an explanation for this relationship is provided in Appendix A.

## 4 Estimation procedure

In Section 3, we developed a rigorous framework that allowed us to introduce the localized, scale-specific wavelet canonical coherence. In this section, we propose a well-behaved estimation procedure for quantifying the canonical coherence and corresponding canonical vectors. We start by estimating the local wavelet spectrum (LWS) matrices in equations (9) and (10) in the spirit of [21], given by

$$\widehat{\mathbf{S}}_{j,k} = \sum_{l=1}^{J} A_{jl}^{-1}\tilde{\mathbf{I}}_{l,k}, \text{ where } \tilde{\mathbf{I}}_{l,k} = \frac{1}{2M+1}\sum_{m=-M}^{M} \mathbf{I}_{l,k+m} \text{ is the smoothed periodogram}, \quad (14)$$

$k$ represents the shift of the wavelet function and is equivalent to time $k = [uT]$ in our context, and $M$ is the half-width of the rectangular smoothing kernel, controlling the amount of temporal smoothing. The matrix $\mathbf{I}_{l,k}$ is the raw periodogram at scale $l$ and time $k$, obtained as

$$\mathbf{I}_{l,k} = \mathbf{d}_{l,k}\mathbf{d}_{l,k}^\top, \text{ where } \mathbf{d}_{l,k} = \sum_{t=0}^{T} \mathbf{X}_t \psi_{l,k}(t) \text{ is the empirical wavelet coefficient vector.}$$

This multistep procedure yields consistent estimators of the LWS matrices under the asymptotic conditions $T, M \to \infty$ and $M/T \to 0$ [21]. We propose the following estimator for the scale-specific wavelet canonical coherence (WaveCanCoh)

$$\widehat{\boldsymbol{\rho}}_{j;\mathbf{XY}}(u) = \widehat{\Lambda}_j^{(1)}(u), \qquad (15)$$

where $\widehat{\Lambda}_j^{(1)}(u)$ is the largest eigenvalue of $\widehat{\mathbf{M}}_{j;\mathbf{a}}(u)$ and $\widehat{\mathbf{M}}_{j;\mathbf{b}}(u)$, defined as

$$\widehat{\mathbf{M}}_{j;\mathbf{a}}(u) = \widehat{\mathbf{S}}_{j,\mathbf{XX}}^{-1}(u)\widehat{\mathbf{S}}_{j,\mathbf{XY}}(u)\widehat{\mathbf{S}}_{j,\mathbf{YY}}^{-1}(u)\widehat{\mathbf{S}}_{j,\mathbf{YX}}(u), \qquad (16)$$

$$\widehat{\mathbf{M}}_{j;\mathbf{b}}(u) = \widehat{\mathbf{S}}_{j,\mathbf{YY}}^{-1}(u)\widehat{\mathbf{S}}_{j,\mathbf{YX}}(u)\widehat{\mathbf{S}}_{j,\mathbf{XX}}^{-1}(u)\widehat{\mathbf{S}}_{j,\mathbf{XY}}(u). \qquad (17)$$

The estimated localized, scale-specific canonical direction vectors $\widehat{\mathbf{a}}_j(u)$ and $\widehat{\mathbf{b}}_j(u)$ are the eigenvectors of $\widehat{\mathbf{M}}_{j;\mathbf{a}}(u)$ and $\widehat{\mathbf{M}}_{j;\mathbf{b}}(u)$ respectively, associated with $\widehat{\Lambda}_j^{(1)}(u)$. These quantities provide estimates of the time-varying global coherence and the channel-specific contributions at scale $j$. The proposed estimators are consistent with the true quantities they aim to approximate, provided certain asymptotic conditions are met. These include increasing sample size and appropriate smoothing bandwidth, ensuring reliable estimation in the limit. *Proof: See Appendix B.*

**Algorithm 1** summarizes the estimation procedure for WaveCanCoh and its results can be further used to investigate the temporal channel contributions to the global association, at a particular scale.

---
**Algorithm 1** Proposed WaveCanCoh estimation algorithm for nonstationary time series
---

Suppose the observed data are two sets of multivariate locally stationary time series, denoted as $\mathbf{X}_t = \left( X_t^{(1)}, \ldots, X_t^{(P)} \right)^\top$ and $\mathbf{Y}_t = \left( Y_t^{(1)}, \ldots, Y_t^{(Q)} \right)^\top$, observed for $t = \{1, \ldots, T\}$.

**1. Fuse:** fuse the data to a new $(P+Q)$-variate time series denoted as $\mathbf{Z}_t$, with $\mathbf{Z}_t = \left( \mathbf{X}_t^\top, \mathbf{Y}_t^\top \right)^\top$. (This fused representation allows for the joint analysis of the two multivariate processes within a unified framework. The Causal-WaveCanCoh can also be estimated by appropriately incorporating the leading-lag into the existing WaveCanCoh estimation procedure.)

**2. Spectral estimation:** estimate the LWS matrix of $\{\mathbf{Z}_t\}$ using equation (14). Denote the estimator as $\widehat{\mathbf{S}}_{j;\mathbf{ZZ}}(u)$ for any rescaled time $u \in (0, 1)$. The estimated auto- and cross-LWS between $\{\mathbf{X}_t\}$ and $\{\mathbf{Y}_t\}$, denoted as $\widehat{\mathbf{S}}_{j;\mathbf{XX}}(u), \widehat{\mathbf{S}}_{j;\mathbf{YY}}(u), \widehat{\mathbf{S}}_{j;\mathbf{XY}}(u), \widehat{\mathbf{S}}_{j;\mathbf{YX}}(u)$, can be obtained by partitioning $\widehat{\mathbf{S}}_{j;\mathbf{ZZ}}(u)$ into four submatrices as illustrated in equation (7).

**3. Eigendecomposition:** compute the matrices $\widehat{\mathbf{M}}_{j;\mathbf{a}}(u)$ and $\widehat{\mathbf{M}}_{j;\mathbf{b}}(u)$ in equations (16)- (17), then perform their eigendecompositions and obtain their (common) largest eigenvalue $\widehat{\Lambda}_j^{(1)}(u)$. This will serve as the estimated WaveCanCoh in (15), $\widehat{\boldsymbol{\rho}}_{j;\mathbf{XY}}(u) = \widehat{\Lambda}_j^{(1)}(u)$, while its corresponding eigenvectors of $\widehat{\mathbf{M}}_{j;\mathbf{a}}(u)$ and $\widehat{\mathbf{M}}_{j;\mathbf{b}}(u)$ give the canonical direction vectors, $\widehat{\mathbf{a}}_j(u)$ and $\widehat{\mathbf{b}}_j(u)$.

## 5 Simulation study

In this section, we implement the proposed framework using simulated data under two distinct scenarios, one adhering to the MvLSW assumptions underpinning our method, while the other introduces nonstationarity without strictly satisfying the MvLSW assumptions. These setups allow us to validate both the theoretical soundness and empirical performance of the proposed approach, as well as to assess its robustness and practical applicability in real-world scenarios where model assumptions may be violated. To further evaluate performance, we also compare the results with those obtained from the classical Fourier-based canonical coherence approach.

**MvLSW-based simulation.** We generate the multivariate time series $\{\mathbf{Z}_t\}$ from a MvLSW process with $P = 6$, $Q = 4$, observed across $T = 1024$ time points. The process is constructed using non-decimated Haar wavelets, with non-zero spectral structure specified at scale $j = 2$, as detailed in Appendix C.1. We impose a weaker dependence structure between $\{\mathbf{X}_t\}$ and $\{\mathbf{Y}_t\}$ in the interval $0 < u < 0.5$, and a stronger dependence in the interval $0.5 < u < 1$, allowing us to examine the framework's sensitivity to changes in cross-group coherence. Using the process realization (Figure 9) we estimate WaveCanCoh using Algorithm 1. To assess the estimation accuracy and account for variability, we replicate the simulation and estimation process 1000 times. At each time point, we compute the average of the estimated WaveCanCoh across the replicates and construct a 95% Wald confidence interval using the empirical variance. Figure 2 (left) demonstrates that the proposed WaveCanCoh method accurately tracks the true coherence structure and effectively reflects its time-varying nature, while the estimated canonical direction vectors in Figure 3 map the temporal and individual channel heterogeneity in their roles within the multivariate dependence structure.

**Mixture of AR(2)-based processes.** To evaluate the robustness and generality of the proposed framework, we investigate WaveCanCoh using synthetic data generated from a mixture of AR(2) processes. Unlike the MvLSW-based simulations, this setting introduces nonstationary dynamics without any wavelet-based structure, providing a more flexible and realistic test scenario. To the best of our knowledge, WaveCanCoh is the first framework designed to estimate time-varying canonical coherence between two multivariate time series groups. However, to benchmark its performance, we compare it against a method (henceforth referred to as LSP) based on the time-varying Cramér representation [7] and described in Appendix C.2, with canonical coherence estimated via STFT-based localized spectra [1]. We simulate 500 replicates of $\mathbf{X}_t \in \mathbb{R}^4$ and $\mathbf{Y}_t \in \mathbb{R}^3$, with $T = 1024$, each formed by mixing five latent AR(2) sources tuned to neural frequency bands. In the first half, shared gamma $(30 - 50Hz)$ bands induce cross-group coherence, while the second half contains no shared structure (see details in Appendix C.2). Figure 2 (right) illustrates that while both methods detect the existence of coherence in the first half, only WaveCanCoh captures its sharp drop and true behaviour in the second half, thus demonstrating its advantage in identifying transient, localized changes that global Fourier-based methods fail to detect.

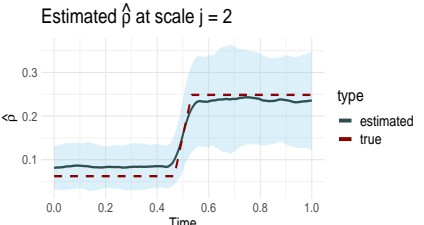
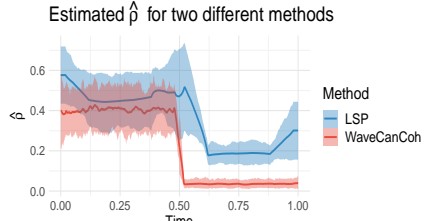

Figure 2: Left: Estimated wavelet canonical coherence at scale $j = 2$ over 1000 MvLSW replications. The solid line shows the average estimated coherence, the dashed line is the true coherence computed from the specified spectrum. Right: Estimated canonical coherence over 500 replicates of the AR(2) mixture using WaveCanCoh and LSP at scale $j = 1$ and $\omega \in [25, 50]Hz$, respectively. Shaded areas indicate the corresponding 95% Wald confidence interval.

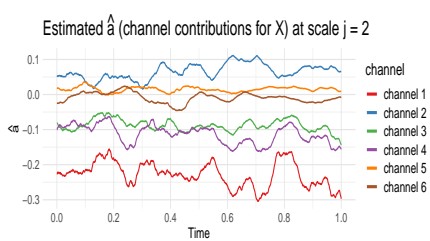
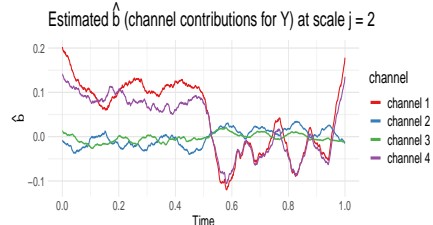

Figure 3: Estimated time-varying canonical direction vectors $\widehat{\mathbf{a}}_2(\cdot)$ (left) and $\widehat{\mathbf{b}}_2(\cdot)$ (right).

## 6 Local field potential (LFP) data analysis

To demonstrate the practical utility of our proposed WaveCanCoh framework, we analyze LFP activity recorded from the hippocampus of rats engaged in a sequence memory task [2, 22]. The data were recorded using a 22-electrode microdrive implanted in the CA1 subregion to capture high-resolution LFP signals across all channels at a sampling rate of $1000Hz$. In this task, rats were tested on their memory of a sequence of five odors (odors ABCDE). Each odor was presented for ∼1.2 second ($s$) and a variable delay of ∼5$s$ separated each odor (see Figure 1). For each trial (i.e., each odor presentation), the rat had to judge whether the odor was presented "in sequence" (e.g., ABC...) or "out of sequence" (e.g., ABD...) and indicate their decision by holding their nosepoke response until a tone signal (at $1.2s$) or withdrawing before the signal, respectively. Correct-response trials (i.e., correct "in sequence" or "out of sequence" decisions) were rewarded. LFP activity data were recorded over a 4$s$ period ($T = 4000$ time points) per trial, with $t = 0$ marking the moment the rat initiated a nosepoke to receive the odor stimulus. This paradigm provides a well-controlled setting to investigate dynamic, time-varying functional interactions in the hippocampus during memory-guided decisions. We employ WaveCanCoh framework with Haar wavelets to quantify frequency-specific functional coherence between two groups of hippocampal electrodes (T1, T2, T4, T5 and T13–T17), and to examine how coherence patterns differ between correct- and incorrect-response trials ("in sequence" trials only). Specifically, we analyze LFP data from the rat Mitt, which included 40 correct-response trials and 32 incorrect-response trials. Figure 4 presents the estimated wavelet canonical coherence at scale $j = 5$, corresponding to the $15.625 - 31.25Hz$ frequency band. The results, averaged across trials for each condition, reveal dynamic changes in inter-regional coherence, with a pronounced peak around the time of odor stimulus delivery ($t = 0$). Notably, distinct patterns emerge between correct and incorrect trials, suggesting that coherent activity in this frequency band may play a role in supporting successful memory retrieval and decision making. More results for several other scales can be found in Figure 10 in Appendix D.1.

To further interpret the coherence patterns, Figure 5 provides a spatial summary of the canonical coherence between the two electrode groups at several selected time points. The double-headed arrows represent the magnitude of estimated coherence between the two regions, while the numbers

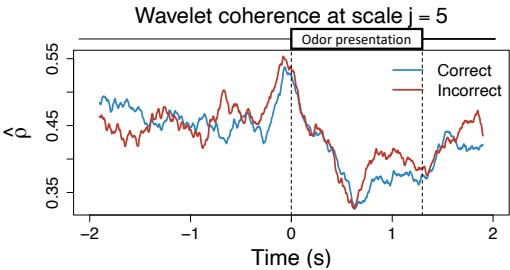

Figure 4: Estimated wavelet canonical coherence between two hippocampal subregions (T1, T2, T4, T5 vs. T13–T17) in one subject (Mitt) at scale $j = 5\,(15.625\text{–}31.25Hz)$. The estimates are averaged across 40 correct- and 32 incorrect-response trials, using a rectangular smoothing window of $0.2s$.

in the circles reflect the individual channel contributions to the global coherence, derived from the elements of the canonical vectors $\mathbf{a}_5(\cdot)$ and $\mathbf{b}_5(\cdot)$.

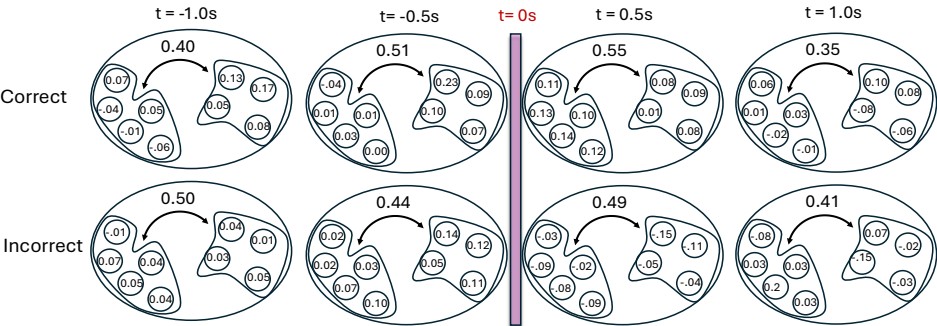

Figure 5: Spatio-temporal illustration of WaveCanCoh between hippocampal regions (T1, T2, T4, T5 and T13–T17) in rat Mitt at scale $j = 5\,(15.625\text{–}31.25Hz)$. Arrows indicate the magnitude of inter-region coherence at selected time points. Numbers inside circles represent the channel-wise relative contributions to the canonical coherence for each region, for correct- and incorrect-responses. Negative values indicate that the corresponding channels contribute to coherence in the negatively correlated direction between the two regions.

The results highlight that both the strength and structure of inter-regional coherence vary dynamically over time and differ across trial outcomes, reflecting the nonstationary nature of neural interactions during task performance. Specifically, incorrect-response trials exhibit lower coherence at time points immediately following the odor stimulus delivery and are often driven by a few dominant channels, while correct trials show higher coherence values and relatively balanced contributions across channels. These findings illustrate the necessity of a framework like WaveCanCoh, which simultaneously identifies time-varying and scale-specific dependence structures between multivariate time series. Traditional stationary or pairwise approaches would fail to capture such nuanced dynamics, highlighting the need for a method like WaveCanCoh to extract such complex relationships in brain activity. The directed interactions between brain regions are also explored in Appendix E with the proposed Causal-WaveCanCoh framework.

To further validate the existence of significant differences in the activity between correct- and incorrect-response trials, we propose a time-specific detection procedure based on the permutation test to determine temporally localized differences in the wavelet canonical coherence at a given scale between conditions, while maintaining the nonparametric nature of the statistical inference. The detailed inference steps are shown in **Algorithm 2** (Appendix D.2). Table 1 reports the permutation test results on the LFP data, with the value in each cell representing the difference in the median wavelet coherence between correct and incorrect trials at the corresponding scale $j$ and time $t^* = -1.0$, $-0.5$, $0.5$, and $1.0$ seconds. The values in parentheses denote the permutation $p$-values obtained using the windowed test procedure (window size $= 0.2s$, $n_{\text{perm}} = 1000$). Detailed test results are shown in Figure 11 (Appendix D.3), revealing significant differences between correct- and incorrect-

response trials at scales $j = 4$ to 7 following the odor stimulation. Statistically significant differences in canonical coherence between correct and incorrect trials emerge at time points following odor sampling ($t = 0$), predominantly at intermediate wavelet scales corresponding approximately to the 8–62 $Hz$ frequency range. On the other hand, no significant differences are observed prior to stimulus onset, indicating that the coherence patterns distinguishing inter-regional communication among trial types are tightly linked to task engagement. In comparison, we implement the LSP algorithm on the same data and conduct the same permutation test. The results (see Table 2) show that this approach lacks the sensitivity to distinguish between correct and incorrect trials. A likely explanation is that the smoothed approximation may have masked the true differences. These findings demonstrate the effectiveness of the proposed WaveCanCoh framework and associated permutation test in capturing localized, frequency-specific differences in neural coordination between behavioral conditions, thus offering a powerful tool for analyzing complex brain interactions.

Table 1: Differences in median wavelet canonical coherence (WaveCanCoh) between correct- and incorrect-response trials across time points and scales. Each cell reports the median difference at scale $j$ and time $t^*$, with $p$-values obtained from the time-specific permutation test shown in parentheses.

| $j$ \ $t^*(s)$ | -1.0 | -0.5 | 0.5 | 1.0 |
|---|---|---|---|---|
| 3 $(62.5 - 125Hz)$ | 0.222 (0.786) | 0.068 (0.704) | -0.178 (0.355) | 0.106 (0.886) |
| 4 $(31.25 - 62.5Hz)$ | -0.090 (0.180) | -0.021 (0.506) | 0.023 (0.001**) | 0.027 (0.691) |
| 5 $(15.63 - 31.25Hz)$ | 0.006 (0.977) | 0.229 (0.999) | 0.334 (0.002**) | 0.016 (0.079) |
| 6 $(7.81 - 15.63Hz)$ | -0.025 (0.239) | -0.049 (0.111) | 0.039 (0.001**) | 0.002 (0.025**) |
| 7 $(< 7.81Hz)$ | -0.014 (0.999) | 0.058 (0.640) | 0.012 (0.059) | -0.030 (0.489) |

Table 2: Differences in median Fourier-based canonical coherence (LSP) between correct- and incorrect-response trials across time points and scales. Each cell reports the median difference at scale $j$ and time $t^*$, with $p$-values obtained from the time-specific permutation test shown in parentheses.

| $j$ \ $t^*(s)$ | -1.0 | -0.5 | 0.5 | 1.0 |
|---|---|---|---|---|
| 3 $(62.5–125\,Hz)$ | -0.065 (0.886) | -0.002 (0.901) | -0.009 (0.991) | -0.060 (0.471) |
| 4 $(31.25–62.5\,Hz)$ | 0.001 (0.128) | -0.044 (0.141) | 0.010 (0.056*) | 0.010 (0.991) |
| 5 $(15.63–31.25\,Hz)$ | 0.009 (0.470) | 0.003 (0.970) | 0.008 (0.999) | 0.003 (0.983) |
| 6 $(7.81–15.63\,Hz)$ | -0.001 (0.512) | -0.002 (0.052*) | 0.002 (0.901) | 0.000 (0.842) |
| 7 $(< 7.81\,Hz)$ | 0.001 (0.094) | 0.001 (0.121) | -0.004 (0.754) | -0.002 (0.901) |

# 7   Conclusions

We introduced a novel methodological framework, scale-specific wavelet canonical coherence (WaveCanCoh), designed to quantify the dynamic multiscale coherence between two sets of nonstationary multivariate time series. Our primary contributions include the rigorous definition of WaveCanCoh within the multivariate locally stationary wavelet framework and the development of a comprehensive estimation and theoretically-backed inference procedure based on wavelet analysis. We validated our proposed methodology through simulation studies, demonstrating its accuracy in tracking true coherence structures. The application to local field potential activity recorded from subregions of the hippocampus effectively showcased the WaveCanCoh capability to identify nuanced spatio-temporal coherence patterns associated with cognitive performance, while our permutation-based inference procedure provided a robust, nonparametric approach for detecting significant coherence differences between conditions. Compared to existing stationary and Fourier-based canonical coherence methods, WaveCanCoh is shown to offer significant advantages, particularly through its ability to adaptively capture transient and time-localized interactions, rendering it highly suitable for analyzing signals in neuroscience and other fields with data exhibiting dynamic cross-group interactions.

## Acknowledgments and Disclosure of Funding

Knight gratefully acknowledges support from UKRI EPSRC NeST Programme Grant EP/X002195/1. Fortin gratefully acknowledges support from NIH (R01-MH115697 and R01-DC017687), NSF (CAREER IOS-1150292, BCS-1439267, DGE-1839285, and NCS-FR-2319618), and the Whitehall Foundation (2010-05-84). Ombao gratefully acknowledges support from KAUST research fund. We sincerely appreciate the time and effort that the anonymous reviewers and area chairs devoted to evaluating our work and are grateful for their valuable, insightful feedback.

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

## Appendix

## A    Basic introduction to wavelets

Wavelet analysis provides a powerful framework for studying signals with both time- and scale-varying structure, making it particularly well-suited for nonstationary data. Unlike traditional Fourier-based methods, which decompose signals into global sinusoidal bases and therefore assume stationarity, wavelets enable localized, adaptive decompositions by projecting signals onto functions that are compact in both time and scale. This localization is achieved through two core operations: scaling, which adjusts the width of the wavelet to analyze different resolution levels, and shifting, which moves the wavelet across time to detect when features occur. Specifically, the wavelet functions at scale $j$ and shift $k$, denoted by $\psi_{j,k}$, are derived from a mother wavelet $\psi$ and defined as

$$\psi_{j,k}(t) = 2^{-j/2}\psi\left(\frac{t - 2^j k}{2^j}\right), \quad j = 1, \ldots, J,$$

where $J$ represents the number of scales. Smaller scale $j$ corresponds to finer (high-frequency) resolution, and larger $j$ captures coarser (low-frequency) trends. A similar construction applies to the father wavelet, denoted by $\phi_{j,k}$, which serves as a scaling function (see [8] and [18] for more details on wavelets). Figure 6 illustrates the effect of scaling and shifting operations on the wavelet function. This ability to isolate both short-lived and long-term features distinguishes wavelet methods from Fourier analysis, allowing for nuanced investigation of nonstationary signals such as neural activity, where structure evolves dynamically across time and scale.

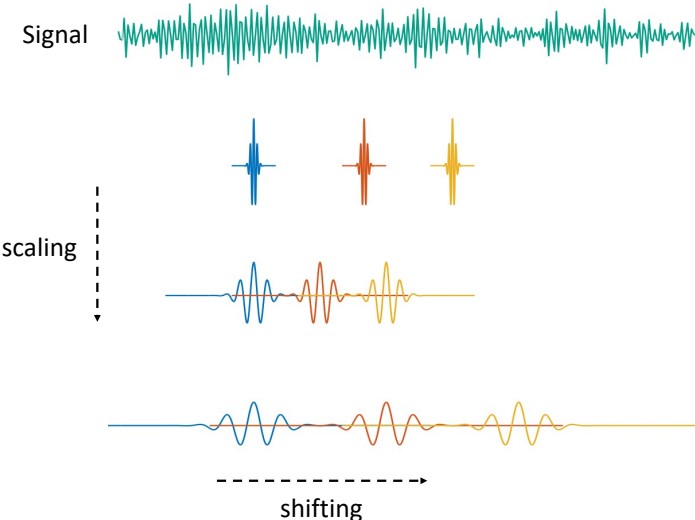

Figure 6: Illustration of scaling and shifting of wavelet functions. Smaller scales (top) capture high-frequency, localized features, while larger scales (bottom) capture broader, low-frequency structures.

The correspondence between wavelet scale and signal frequency arises from the principle of multi-resolution analysis, which provides the foundational framework for wavelet construction and signal representation. Briefly, the discrete wavelet transform (DWT) decomposes a signal into frequency subbands via a dyadic filter bank architecture. At each level, the signal is passed through a pair of conjugate quadrature filters: a low-pass filter $h[n]$ and a high-pass filter $g[n]$, followed by downsampling by a factor of two. The low-pass branch yields approximation coefficients that retain coarse-scale information, while the high-pass branch produces detail coefficients that capture localized high-frequency variations. Specifically, given a discrete signal $x[n]$, the approximation and detail coefficients at level $j$ are obtained by

$$a_j[n] = \sum_k h[k]a_{j-1}[2n - k], \quad d_j[n] = \sum_k g[k]a_{j-1}[2n - k],$$

where $a_{j-1}$ is the approximation from the previous level (with $a_0 = x$). This process is iterated on the low-pass output, producing a multiscale representation in which each level isolates a specific frequency band. For a signal sampled at rate $f_s$, the detail coefficients at level $j$ correspond approximately to the frequency interval $\left[\frac{f_s}{2^{j+1}}, \frac{f_s}{2^j}\right]$. The orthogonality between subbands ensures perfect reconstruction and energy preservation, and the hierarchical filter bank provides a localized time-frequency analysis with increasingly coarse temporal resolution at lower frequencies. The explanation above offers an intuitive understanding of how components at each wavelet scale can be approximated to true frequency bands (see Figure 7 for an intuitive illustration). This approximation enables our framework to capture the overall association between two sets of locally stationary time series in both the temporal and frequency domains (more details can be found in [8]).

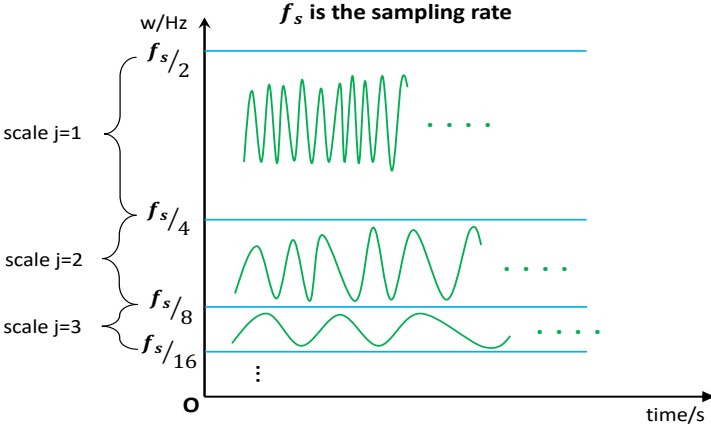

Figure 7: Illustration of mapping relationship between wavelet scales and their approximated frequency bands.

## B    Theoretical proofs

**Proof of solution to equation (8).** Suppose that $\mathbf{X}_t$ and $\mathbf{Y}_t$ are two multivariate time series and for each time point $t$, the goal is to find the vectors $\mathbf{a}_j(u)$ and $\mathbf{b}_j(u)$ that maximize the coherence,

$$\rho_{j;\mathbf{XY}}(u) = \left\{\mathbf{a}_j^\top(u)\mathbf{S}_{j;\mathbf{XY}}(u)\mathbf{b}_j(u)\right\}^2$$

subject to the normalization constraints described in the main paper

$$\mathbf{a}_j^\top(u)\mathbf{S}_{j;\mathbf{XX}}(u)\mathbf{a}_j(u) = 1, \quad \mathbf{b}_j^\top(u)\mathbf{S}_{j;\mathbf{YY}}(u)\mathbf{b}_j(u) = 1.$$

Assuming rescaled time $u$ and scale $j$ are fixed, we suppress them for clarity. To solve the above optimization problem, we set up the Lagrangian as

$$\mathcal{L}(\mathbf{a}, \mathbf{b}, \lambda_1, \lambda_2) = \frac{1}{2}\left(\mathbf{a}^\top\mathbf{S}_{XY}\mathbf{b}\right)^2 - \frac{\lambda_1}{2}\left(\mathbf{a}^\top\mathbf{S}_{XX}\mathbf{a} - 1\right) - \frac{\lambda_2}{2}\left(\mathbf{b}^\top\mathbf{S}_{YY}\mathbf{b} - 1\right)$$

Differentiating the Lagrangian with respect to $\mathbf{a}$ and $\mathbf{b}$, respectively and requiring the partial derivatives to equal zero, we obtain

$$\frac{\partial\mathcal{L}}{\partial\mathbf{a}} = (\mathbf{S}_{XY}\mathbf{b}) \cdot (\mathbf{S}_{XY}\mathbf{b})^\top \mathbf{a} - \lambda_1\mathbf{S}_{XX}\mathbf{a} = \mathbf{0}$$

$$\Rightarrow (\mathbf{S}_{XY}\mathbf{b}) \cdot (\mathbf{S}_{XY}\mathbf{b})^\top \mathbf{a} = \lambda_1\mathbf{S}_{XX}\mathbf{a}, \qquad \text{(i)}$$

$$\frac{\partial\mathcal{L}}{\partial\mathbf{b}} = (\mathbf{S}_{YX}\mathbf{a}) \cdot (\mathbf{S}_{YX}\mathbf{a})^\top \mathbf{b} - \lambda_2\mathbf{S}_{YY}\mathbf{b} = \mathbf{0}$$

$$\Rightarrow (\mathbf{S}_{YX}\mathbf{a}) \cdot (\mathbf{S}_{YX}\mathbf{a})^\top \mathbf{b} = \lambda_2\mathbf{S}_{YY}\mathbf{b}. \qquad \text{(ii)}$$

With (i), (ii) and since $\left(\mathbf{a}^\top \mathbf{S}_{XY}\mathbf{b}\right)$ is a real-valued quantity, we further obtain

$$(\mathbf{a}^\top \mathbf{S}_{XY}\mathbf{b}) \cdot (\mathbf{a}^\top \mathbf{S}_{XY}\mathbf{b}) = \lambda_1 \mathbf{a}^\top \mathbf{S}_{XX}\mathbf{a}, \text{ and}$$
$$(\mathbf{b}^\top \mathbf{S}_{YX}\mathbf{a}) \cdot (\mathbf{a}^\top \mathbf{S}_{XY}\mathbf{b}) = \lambda_2 \mathbf{b}^\top \mathbf{S}_{YY}\mathbf{b}.$$

Recalling the constraints $\mathbf{a}^\top \mathbf{S}_{XX}\mathbf{a} = 1$ and $\mathbf{b}^\top \mathbf{S}_{YY}\mathbf{b} = 1$, it immediately follows that

$$\lambda_1 = \lambda_2 = \left(\mathbf{a}^\top \mathbf{S}_{XY}\mathbf{b}\right)^2 := \lambda.$$

By substituting the above back into (i), (ii) and assuming $\lambda$ to be non-zero, we obtain

$$\mathbf{a} = \tfrac{1}{\sqrt{\lambda}}\mathbf{S}_{XX}^{-1}\mathbf{S}_{XY}\mathbf{b} \text{ and}$$
$$\mathbf{b} = \tfrac{1}{\sqrt{\lambda}}\mathbf{S}_{YY}^{-1}\mathbf{S}_{YX}\mathbf{a},$$

which plugged into (ii), (i), respectively, yield

$$\mathbf{S}_{XY}\mathbf{S}_{YY}^{-1}\mathbf{S}_{YX}\,\mathbf{a} = \lambda\mathbf{S}_{XX}\mathbf{a} \qquad\qquad\qquad \mathbf{S}_{XX}^{-1}\mathbf{S}_{XY}\mathbf{S}_{YY}^{-1}\mathbf{S}_{YX}\mathbf{a} = \lambda\mathbf{a},$$
$$\mathbf{S}_{YX}\mathbf{S}_{XX}^{-1}\mathbf{S}_{XY}\mathbf{b} = \lambda\mathbf{S}_{YY}\mathbf{b} \text{ or, equivalently,} \qquad \mathbf{S}_{YY}^{-1}\mathbf{S}_{YX}\mathbf{S}_{XX}^{-1}\mathbf{S}_{XY}\mathbf{b} = \lambda\mathbf{b}.$$

Hence $\lambda = \left(\mathbf{a}^\top \mathbf{S}_{XY}\mathbf{b}\right)^2$ is an eigenvalue for both matrices

$$\mathbf{S}_{XX}^{-1}\mathbf{S}_{XY}\mathbf{S}_{YY}^{-1}\mathbf{S}_{YX}, \text{ and}$$
$$\mathbf{S}_{YY}^{-1}\mathbf{S}_{YX}\mathbf{S}_{XX}^{-1}\mathbf{S}_{XY}$$

whose corresponding eigenvectors are $\mathbf{a}$ and $\mathbf{b}$, respectively. Thus, the defined canonical coherence $\rho = \max\left(\mathbf{a}^\top \mathbf{S}_{XY}\mathbf{b}\right)^2$ is the largest eigenvalue of above matrices. The case when $\lambda = 0$ illustrates the canonical coherence between $\mathbf{X}$ and $\mathbf{Y}$ is 0, which is not a meaningful scenario for this problem. We recall the above equations hold for every time $u$ and scale $j$.

**Proof of consistency of WaveCanCoh estimator.** We aim to establish the consistency of the matrix estimators

$$\widehat{\mathbf{M}}_{j;\mathbf{a}} = \widehat{\mathbf{S}}_{j,\mathbf{XX}}^{-1}\widehat{\mathbf{S}}_{j,\mathbf{XY}}\widehat{\mathbf{S}}_{j,\mathbf{YY}}^{-1}\widehat{\mathbf{S}}_{j,\mathbf{YX}}, \quad \widehat{\mathbf{M}}_{j;\mathbf{b}} = \widehat{\mathbf{S}}_{j,\mathbf{YY}}^{-1}\widehat{\mathbf{S}}_{j,\mathbf{YX}}\widehat{\mathbf{S}}_{j,\mathbf{XX}}^{-1}\widehat{\mathbf{S}}_{j,\mathbf{XY}}$$

According to [19] and [21], the smoothed periodogram-based estimators of the local wavelet spectral (LWS) matrices are consistent. Specifically, as the number of time points $T \to \infty$ and the smoothing parameter $M \to \infty$ with $M/T \to 0$, we have

$$\widehat{\mathbf{S}}_{j,\mathbf{XX}} \xrightarrow{P} \mathbf{S}_{j,\mathbf{XX}}, \quad \widehat{\mathbf{S}}_{j,\mathbf{XY}} \xrightarrow{P} \mathbf{S}_{j,\mathbf{XY}}, \quad \widehat{\mathbf{S}}_{j,\mathbf{YX}} \xrightarrow{P} \mathbf{S}_{j,\mathbf{YX}}, \quad \widehat{\mathbf{S}}_{j,\mathbf{YY}} \xrightarrow{P} \mathbf{S}_{j,\mathbf{YY}}.$$

Assuming the spectral matrices and their estimators are non-singular, it follows by the continuous mapping theorem that

$$\widehat{\mathbf{S}}_{j,\mathbf{XX}}^{-1} \xrightarrow{P} \mathbf{S}_{j,\mathbf{XX}}^{-1}, \quad \widehat{\mathbf{S}}_{j,\mathbf{YY}}^{-1} \xrightarrow{P} \mathbf{S}_{j,\mathbf{YY}}^{-1}.$$

Since matrix multiplication is continuous with respect to convergence in probability, we obtain the consistency of the matrix estimators by Slutsky's theorem, namely

$$\widehat{\mathbf{M}}_{j;\mathbf{a}} \xrightarrow{P} \mathbf{M}_{j;\mathbf{a}}, \quad \widehat{\mathbf{M}}_{j;\mathbf{b}} \xrightarrow{P} \mathbf{M}_{j;\mathbf{b}}.$$

Consequently, the estimated wavelet canonical coherence and associated canonical vectors, derived from the largest eigenvalue and corresponding eigenvectors of $\widehat{\mathbf{M}}_{j;a}$ and $\widehat{\mathbf{M}}_{j;b}$, also converge in probability to their population counterparts, following arguments akin to those in [16].

## C  Details of the simulation setup

This appendix provides the full specification of the simulation experiments described in Section 5 of the main text. The code is provided in the supplementary materials for results reproducibility, and here we firstly provide a brief discussion on the computational complexity of the method. The estimation of the spectrum step leads to a time complexity $\mathcal{O}\left(JT(P+Q)^2\right)$ (where the number of scales $J$ is most commonly used as $J = \log_2(T)$). The eigenvalue decomposition step takes

the total time complexity to $\mathcal{O}(JTd^3)$ (let $P \approx Q = d$), to be compared to that of the standard CCA, $\mathcal{O}(Td^2 + d^3)$. The proposed algorithm is naturally more expensive than standard CCA, since it is designed to obtain time-localized and scale-specific results. In our practical experience with WaveCanCoh, the spectral estimation step is extremely fast, with virtually all of the computational time being spent on the eigenvalue decomposition step. Although compared to standard CCA, our method produces a set of results at each time point, resulting in increased computational burden, in the experiments with $T = 1024$, a single replicate is completed within 2.5 seconds on a standard personal computer (Apple Mac, 16GB RAM, 6-core CPU) without resorting to parallel computing or to the use of a cluster, and all results reported in the paper can be obtained within 3 hours. Moreover, computations for WaveCanCoh can be streamlined by storing canonical vectors and coherence values at each time and scale, with the requirement being $\mathcal{O}(JTd^2)$, and the memory complexity $\mathcal{O}(d^2)$ for standard, one time point global estimate.

### C.1 MvLSW-based simulation

We simulate a $P + Q = 6 + 4$ dimensional multivariate time series $\mathbf{Z}_t = [\mathbf{X}_t^\top, \mathbf{Y}_t^\top]^\top \in \mathbb{R}^{10}$ from a multivariate locally stationary wavelet (MvLSW) process over $T = 1024$ time points. The wavelet spectrum $\mathbf{S}_{j;\mathbf{ZZ}}(u)$ is non-zero only at scale $j = 2$ and is structured as a $10 \times 10$ block matrix

$$\mathbf{S}_{j=2;\mathbf{ZZ}}(u) = \begin{bmatrix} \mathbf{S}_{j;\mathbf{XX}} & \mathbf{S}_{j;\mathbf{XY}}(u) \\ \mathbf{S}_{j;\mathbf{YX}}(u) & \mathbf{S}_{j;\mathbf{YY}} \end{bmatrix}, \quad u = \frac{t}{T} \in (0,1).$$

**Auto-spectral block:** $\mathbf{S}_{j;\mathbf{XX}} \in \mathbb{R}^{6\times6}$

$$\mathbf{S}_{j;\mathbf{XX}} = \begin{bmatrix} 8 & 1 & 1 & 0 & 0 & 0 \\ 1 & 8 & 0 & 0 & 0 & 1 \\ 1 & 0 & 8 & 0 & 0 & 0 \\ 0 & 0 & 0 & 8 & 1 & 0 \\ 0 & 0 & 0 & 1 & 8 & 0 \\ 0 & 1 & 0 & 0 & 0 & 8 \end{bmatrix}$$

**Auto-spectral block:** $\mathbf{S}_{j;\mathbf{YY}} \in \mathbb{R}^{4\times4}$

$$\mathbf{S}_{j;\mathbf{YY}} = \begin{bmatrix} 6 & 0 & 1 & 0 \\ 0 & 6 & 1 & 1 \\ 1 & 1 & 6 & 0 \\ 0 & 1 & 0 & 6 \end{bmatrix}$$

**Cross-spectral block:** $\mathbf{S}_{j;\mathbf{XY}}(u) \in \mathbb{R}^{6\times4}$  This block is time-varying. For $u < 0.5$, each specified cross-group pair has spectrum value 1; for $u \geq 0.5$, the same entries increase to 2.

Let

$$c(u) = \begin{cases} 1 & \text{if } u < 0.5 \\ 2 & \text{if } u \geq 0.5 \end{cases}$$

then

$$\mathbf{S}_{j;\mathbf{XY}}(u) = \begin{bmatrix} c(u) & 0 & 0 & c(u) \\ 0 & c(u) & 0 & 0 \\ 0 & 0 & 0 & c(u) \\ c(u) & 0 & 0 & 0 \\ 0 & 0 & 0 & 0 \\ 0 & 0 & 0 & 0 \end{bmatrix}, \quad \mathbf{S}_{j;\mathbf{YX}}(u) = \mathbf{S}_{j;\mathbf{XY}}^\top(u)$$

The entries of the cross-spectral matrix determine how individual channels contribute to the global coherence structure between $\{\mathbf{X}_t\}$ and $\{\mathbf{Y}_t\}$. Figure 8 visualizes the spectral structure $\mathbf{S}_{j=2;\mathbf{ZZ}}(u)$, and Figure 9 shows example realizations of the simulated processes $\{\mathbf{X}_t\}$ and $\{\mathbf{Y}_t\}$.

### C.2 Mixture of AR(2)-based simulation

**Fourier-based LSP method for comparison**  As a benchmark, we implement a method based on the time-varying Cramér representation for locally stationary processes (LSP), introduced in [7]. A

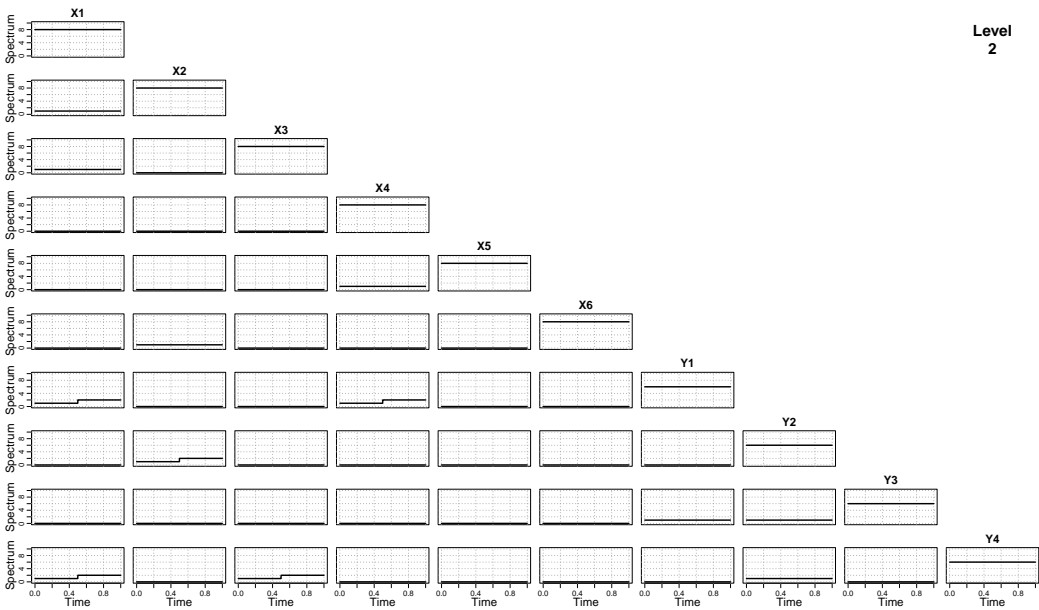

Figure 8: Visualization of the specified block structure of $\mathbf{S}_{2;\mathbf{ZZ}}(u)$.

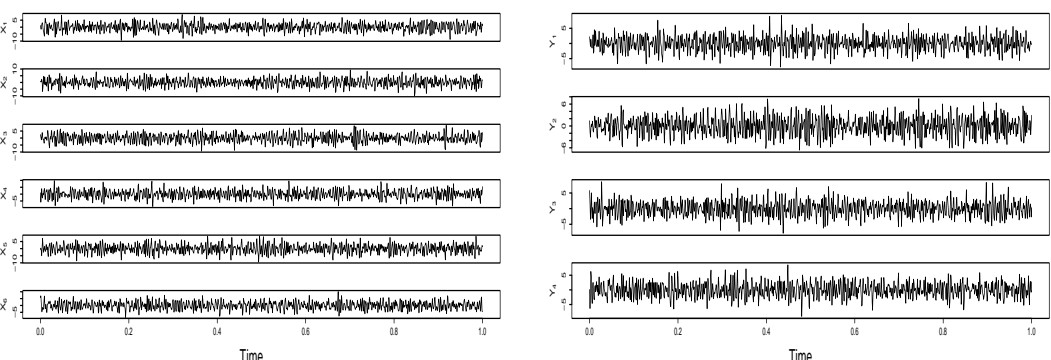

Figure 9: Example realization of $\{\mathbf{X}_t\}$ and $\{\mathbf{Y}_t\}$ generated from stated MvLSW process $\{\mathbf{Z}_t\}$.

locally stationary process $\{\mathbf{X}_{t,T}\}$ can be expressed as

$$\mathbf{X}_{t,T} = \int_{-0.5}^{0.5} \mathbf{A}(u,\omega)e^{2\pi i \omega t}\, dZ(\omega), \quad u = t/T,$$

where $\mathbf{A}(u,\omega)$ is a smoothly varying transfer function and $Z(\omega)$ is a complex orthogonal increment process with $\mathbb{E}[|dZ(\omega)|^2] = d\omega$. The local spectral density is then defined as $\mathbf{f}(u,\omega) = |\mathbf{A}(u,\omega)|^2$. We estimate $\mathbf{f}(u,\omega)$ via the Short-Time Fourier Transform (STFT) with a Gaussian smoothing kernel and compute canonical coherence over $\omega \in [25, 50]Hz$.

**Simulation setting** We simulate 500 independent replicates of two multivariate processes, $\{\mathbf{X}_t\} \in \mathbb{R}^4$ and $\{\mathbf{Y}_t\} \in \mathbb{R}^3$, for $t = 1, \ldots, T$, with $T = 1024$ and sampling rate $f_s = 100Hz$. Each process is generated from mixture of $K = 5$ latent AR(2) sources, $\mathbf{Z}_t^{(\mathbf{X})}, \mathbf{Z}_t^{(\mathbf{Y})} \in \mathbb{R}^K$, peaking at different frequency bands (delta, theta, alpha, beta, gamma), respectively. For the first half of the time series ($t \leq T/2$), the observed processes are given by:

$$\mathbf{X}_t = B^{(1)}\mathbf{Z}_t^{(\mathbf{X})}, \qquad \mathbf{Y}_t = C^{(1)}\mathbf{Z}_t^{(\mathbf{Y})},$$

and for the second half ($t > T/2$):

$$\mathbf{X}_t = B^{(2)}\mathbf{Z}_t^{(\mathbf{X})}, \qquad \mathbf{Y}_t = C^{(2)}\mathbf{Z}_t^{(\mathbf{Y})},$$

where the mixing matrices are:

$$B^{(1)} = \begin{bmatrix} 0 & 0 & 0 & 0 & 0.95 \\ 0 & 0 & 0 & 0 & 0.90 \\ b_{3,1}^{(1)} & b_{3,2}^{(1)} & 0 & 0 & 0 \\ b_{4,1}^{(1)} & b_{4,2}^{(1)} & b_{4,3}^{(1)} & 0 & 0 \end{bmatrix}, \quad B^{(2)} = \begin{bmatrix} 0 & b_{1,2}^{(2)} & b_{1,3}^{(2)} & 0 & 0 \\ 0 & 0 & 0.80 & 0 & 0 \\ 0.90 & 0 & 0 & 0 & 0 \\ 0 & b_{4,2}^{(2)} & b_{4,3}^{(2)} & 0 & 0 \end{bmatrix}$$

$$C^{(1)} = \begin{bmatrix} 0 & 0 & 0 & 0 & 0.95 \\ 0 & 0 & 0 & 0 & 0.90 \\ 0 & c_{3,2}^{(1)} & c_{3,3}^{(1)} & 0 & 0 \end{bmatrix}, \quad C^{(2)} = \begin{bmatrix} 0 & 0 & 0 & 0.90 & 0 \\ 0 & 0 & c_{2,3}^{(2)} & 0 & 0 \\ c_{3,1}^{(2)} & 0 & 0 & 0 & 0 \end{bmatrix}.$$

For each row of the mixing matrices, a selected subset of frequency bands is assigned non-zero weights that are randomly drawn to sum to a predefined total (e.g., 0.95, 0.90, or 1.0), introducing controlled variability across replicates while preserving the intended contribution structure. The latent sources $\mathbf{Z}_t^{(\mathbf{X})}$ and $\mathbf{Z}_t^{(\mathbf{Y})}$ are partially shared in the gamma band (component 5) during the first regime, with mixing weights $\alpha = 0.7$ and $\beta = 0.6$, respectively:

$$Z_{t,5}^{(\mathbf{X})} = \alpha Z_{t,5}^{(\text{shared})} + (1-\alpha)Z_{t,5}^{(\mathbf{X},\text{private})}, \qquad Z_{t,5}^{(\mathbf{Y})} = \beta Z_{t,5}^{(\text{shared})} + (1-\beta)Z_{t,5}^{(\mathbf{Y},\text{private})}, \quad t \le T/2$$

where $Z_{t,5}^{(\text{shared})}$ is a common latent process. Specifically, each channel, say $p$, of $\{\mathbf{Z}_t^{(\text{shared})}\}$, $\{\mathbf{Z}_t^{(\mathbf{X},\text{private})}\}$ and $\{\mathbf{Z}_t^{(\mathbf{Y},\text{private})}\}$ is generated from the following AR(2) process independently,

$$Z_{t,p} = \phi_1 Z_{t-1,p} + \phi_2 Z_{t-2,p} + w_t,$$

where $\{w_t\}$ is white noise and the coefficients are $\phi_1 = 2cos(2\pi\eta)/e^s$, $\phi_2 = -1/e^{2s}$. For each component $p = 1, \ldots, 5$, we use the frequency vector, $\boldsymbol{\eta} = \{\eta^{(1)}, \ldots, \eta^{(5)}\} = \{0.02, 0.06, 0.10, 0.175, 0.375\}$, and the sharpness parameter $\mathbf{s} = \{s^{(1)}, \ldots, s^{(5)}\} = \{0.03, 0.03, 0.03, 0.05, 0.05\}$, where smaller $s$ yields narrower frequency bands.

This design induces time-varying coherence between the first two channels of $\mathbf{X}$ and $\mathbf{Y}$ during the first regime, primarily through the shared gamma component, while maintaining independence during the second regime. The abrupt transition in mixing structure at $t = T/2$ provides a controlled setting for evaluating the sensitivity of coherence estimation methods to sudden changes in cross-dependence.

## D    Additional analysis for the LFP data in Section 6

### D.1    Additional results

We display the estimated wavelet canonical coherence between two electrode clusters across additional scales. Figure 10 shows that cross-group associations vary significantly over time, with the stimulus clearly eliciting a scale-dependent response, indicative of the heterogeneous brain activity across different frequency bands. These results emphasize the importance of capturing scale-specific, time-varying coherence.

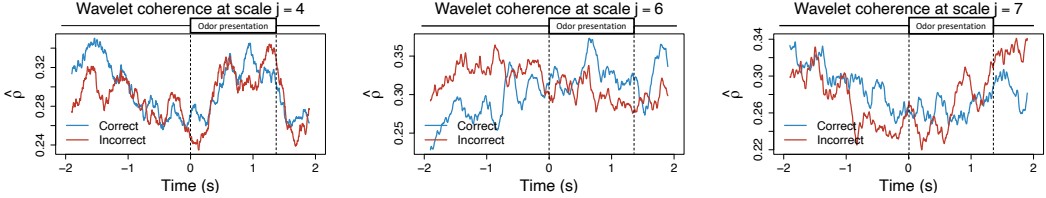

Figure 10: Estimated wavelet canonical coherence between two hippocampal regions (electrodes T1, T2, T4, T5 vs. T13–T17) in rat Mitt at scale $j = 4\,(31.25\text{–}62.5Hz)$, $j = 6\,(7.81\text{–}15.63Hz)$ and $j = 7\,(3.90\text{–}7.81Hz)$. The estimates are averaged across 40 correct- and 32 incorrect-response trials, using a rectangular smoothing window of 0.2 seconds.

### D.2 Permutation test: Algorithm

Algorithm 2 below gives the detailed procedure used in Section 6 for detecting significant differences in canonical coherence between correct- and incorrect-response trials, at selected time points.

---

**Algorithm 2** Time-specific window permutation test for wavelet coherence analysis

Assume the time-localized, scale-specific WaveCanCoh $\rho^{(r)}_{\text{correct/ incorrect}}(j, u)$ is available for each trial $r$, along with the trial label 'correct/ incorrect'. We want to test for significant differences between correct- and incorrect-response trial groups at specific time points $t^*$ and given scale $j$.

**1.Window definition:** For each time point $t^*$, define a window of size $w$ centered at $t^*$:

$$t_{\text{start}} = t^* - w/2, \quad t_{\text{end}} = t^* + w/2$$

**2.Test statistic:** For each time point $t^*$ and given scale $j$, compute:

$$T_{\text{obs}}(j, t^*) = \sum_{t=t_{\text{start}}}^{t_{\text{end}}} \left( \text{median}_r \left( \rho^{(r)}_{\text{correct}}(j, t/T) \right) - \text{median}_r \left( \rho^{(r)}_{\text{incorrect}}(j, t/T) \right) \right)^2$$

**3.Permutation:** For each $t^*$ and scale $j$:
- Combine all $\rho^{(r)}(j, t/T)$ values from both correct- and incorrect-response trials across the time window.
- Perform $n_{\text{perm}}$ random permutations. For each permutation $i$:
  - Randomly assign trials into two new groups of the same sizes as the original groups.
  - Compute the permuted statistic:

  $$T^{(i)}_{\text{perm}}(j, t^*) = \sum_{t=t_{\text{start}}}^{t_{\text{end}}} \left( \text{median}_r \left( \rho^{(r)}_{\text{perm},1}(j, t/T) \right) - \text{median}_r \left( \rho^{(r)}_{\text{perm},2}(j, t/T) \right) \right)^2$$

  where $\rho^{(r)}_{\text{perm},1}$ and $\rho^{(r)}_{\text{perm},2}$ are the permuted trial groups.

**4.Calculate $p$-value:**

$$p(j, t^*) = \frac{1}{n_{\text{perm}}} \sum_{i=1}^{n_{\text{perm}}} \mathbb{I} \left( T^{(i)}_{\text{perm}}(j, t^*) \geq T_{\text{obs}}(j, t^*) \right)$$

---

### D.3 Permutation test: LFP results

Table 1 summarizes permutation test results on the LFP data, where each cell reports the difference in median wavelet coherence between correct and incorrect trials at scale $j$ and time $t^* = -1.0, -0.5,$ $0.5,$ and $1.0s$. Figure 11 shows the full test distributions, highlighting significant differences at scales $j = 4$ to $7$ after odor stimulation.

## E   Causal-WaveCanCoh analysis

To further explore directed interactions between brain regions, we implement the Causal-WaveCanCoh framework (equation (13)) on LFP activity data, which extends WaveCanCoh by introducing a lead-lag structure to evaluate time-lagged canonical coherence. Specifically, we define $\mathbf{X}_t = \left( X^{(\text{T1})}, X^{(\text{T2})}, X^{(\text{T4})}, X^{(\text{T5})} \right)$ and $\mathbf{Y}_t = \left( Y^{(\text{T13})}, \ldots, Y^{(\text{T17})} \right)$ as the multivariate signals corresponding to the two investigated distinct hippocampal regions, and we conduct the analysis in both directions, namely $\mathbf{X}_t \to \mathbf{Y}_{t+h}$ and $\mathbf{Y}_t \to \mathbf{X}_{t+h}$, for lags $h = 0, 10, 20, 30, 40, 50$, corresponding to time shifts from 0 (contemporaneous dependence, used as reference point) to 0.05 seconds.

Figure 12 shows the average scale-specific estimated causal canonical coherence for both directions across the odor presentation time ($0s - 1.2s$) and trial type (correct- vs incorrect-response trials). The results reveal lag-dependent and scale-driven patterns of directed coherence that identify a stronger association between the two hippocampal regions at scale $j = 5$ (corresponding to the frequency band

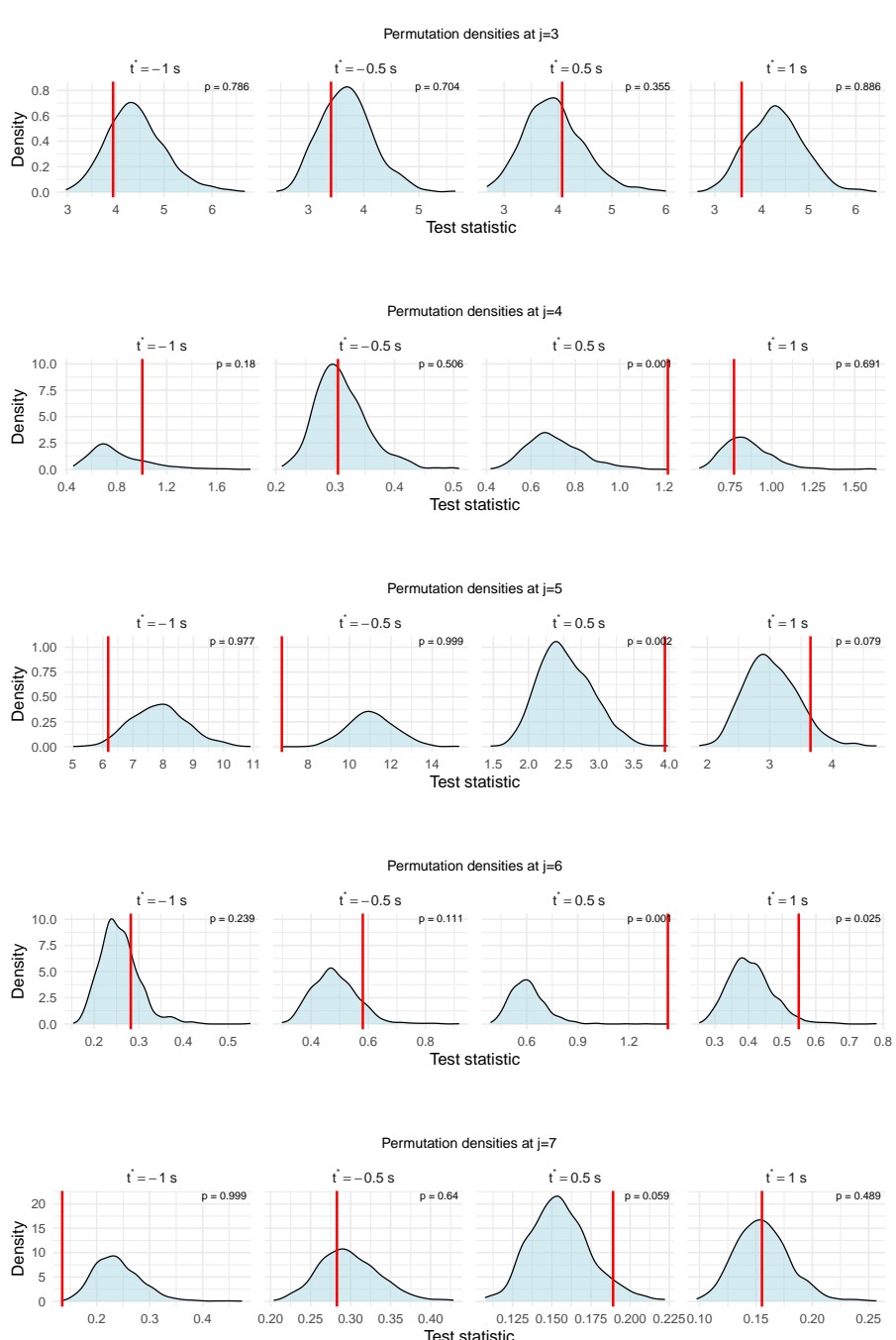

Figure 11: Permutation test distributions ($T_{perm}$) obtained using Algorithm 2 based on WaveCanCoh for different time points across scales $j = 3$ to $j = 7$. Red vertical lines indicate observed test statistics ($T_{obs}$); $p$-values are shown in each panel, corresponding to the results reported in Table 1.

$15.625 - 31.25Hz$). This notably occurs in both directions, with the activity in T13-T17 leading that of T1-T5 in correct-response trials after $h > 10$. The strength and behavior of coherence vary across scales, as well as across correct- and incorrect-response trial groups, which can be captured by our Causal-WaveCanCoh framework.

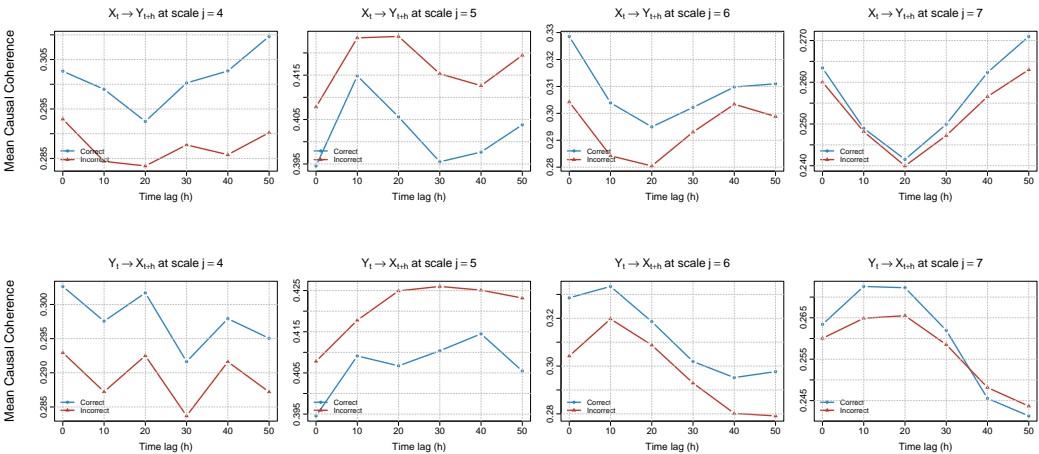

Figure 12: Average causal wavelet canonical coherence across time lags $h = 0$ to 50 at scales $j = 4$ to 7, comparing correct- and incorrect-response trials. Top: $\mathbf{X}_t \to \mathbf{Y}_{t+h}$. Bottom: $\mathbf{Y}_t \to \mathbf{X}_{t+h}$.

## F   Limitations

While our proposed WaveCanCoh framework provides a robust, nonparametric approach for quantifying scale-specific time-varying canonical coherence between two sets of nonstationary multivariate time series, it is inherently limited to a fixed set of wavelet scales. This restricts its flexibility in applications requiring precise frequency localization or alignment with arbitrary bands. As shown in Figure 7, the mapping between scales and true frequency depends on the sampling rate and signal spectrum. For signals with broad frequency content or low sampling rates, certain bands may be poorly resolved, for example, if the sampling rate is $50Hz$, it becomes infeasible to resolve components in the $30 - 40Hz$ range. One practical solution is downsampling when a high sampling rate is available, allowing better alignment between scales and target frequency bands. However, due to time-frequency trade-offs in wavelet analysis, a fully flexible frequency resolution remains challenging. Future work may consider adaptive or overcomplete wavelets to improve frequency targeting while retaining nonstationary modeling capabilities.

