# OpenReview forum: "Wavelet Canonical Coherence for Nonstationary Signals"
_NeurIPS.cc/2025/Conference — NeurIPS 2025 spotlight_

### Official Review · Reviewer_XB1w · 2025-07-02

**Clarity:** 3
**Significance:** 2
**Originality:** 3
**Rating:** 4
**Confidence:** 3

**Summary:**

This paper studies the dependence between two clusters of multivariate time series with canonical coherence in the spectral domain. The authors specifically focus on brain activity signals which are nonstationary. Authors provide experiments and analysis to demonstrate the effectiveness of the proposed work.

**Questions:**

Please see above

**Ethical Concerns:**

["NO or VERY MINOR ethics concerns only"]

**Final Justification:**

Thanks authors for addressing the questions. After reading comments and justifications, I decide to raise final rating.

**Limitations:**

yes

**Quality:**

3

**Strengths And Weaknesses:**

Strengths:
- The topic of studying dependency between multivariate signals is of great interest.
- Authors provide simulation study as supportive evidence for the proposed work.
- The paper is well-organized and easy to follow.

Weaknesses:
- The proposed methodology is not quite innovative in my own opinion. Would be great if author can further emphasize the differences between the proposed method and existing techniques, and highlight the original work.
- The experiment section lacks supportive evidence from more large-scale experiments. It is suggested to do validations on more datasets especially on real data.

---

> ### Author Rebuttal · Authors · 2025-07-30
>
> We sincerely thank you for your thoughtful and detailed comments on our paper，many of your suggestions are highly valuable and have been very helpful in improving the quality of our paper. Please allow us to provide clarifications and responses to the concerns you raised.
> ## Responses for raised weakness
> ### W1. Innovations of WaveCanCoh
> We sincerely thank you for raising the concern regarding the novelty of our proposed method. Below, we clarify the key differences between our proposed WaveCanCoh framework and existing methods, emphasizing its theoretical and practical contributions beyond prior wavelet- or CCA-based approaches.
>
> 1. Beyond Traditional Canonical Coherence Methods.
>
> Classical canonical coherence analysis (CCA), including its spectral domain formulation [Brillinger, 2001], has been extensively studied under the assumption of stationarity. These approaches inherently disregard temporal variation in the dependence structure, which is particularly limiting in domains like neuroscience or finance where interactions evolve over time. WaveCanCoh directly addresses this limitation by modeling {\em time-varying} canonical coherence between two sets of nonstationary multivariate time series. It achieves this by embedding CCA in the multivariate locally stationary wavelet (MvLSW) framework, thus subsequently offering scale-specific and temporally localized inference.
>
> 2. Distinction from Existing Wavelet-Based Dependence Measures.
>
> While wavelet coherence (e.g., Grinsted et al., 2004) has been widely used for pairwise time-frequency analysis, existing work does not extend this to between-group canonical coherence. Previous wavelet-based methods are typically limited to univariate or bivariate signal analysis and do not generalize to multivariate set-to-set dependence. To the best of our knowledge, WaveCanCoh is the first method to combine wavelet-domain multivariate analysis with canonical coherence, enabling estimation of scale-specific, time-varying dependence between signal groups.
>
> 3. Advantages Over Fourier-Based CCA Extensions.
>
> Some studies have extended canonical coherence to the spectral domain via Fourier-based techniques. However, such approaches rely on global basis functions (sinusoids), which are not localized in time, making them unsuitable for detecting transient phenomena. Our WaveCanCoh, grounded in compactly supported wavelets, captures local, frequency-specific dependencies with high temporal resolution, which is clearly demonstrated in both simulation and LFP applications (e.g., Figures 2–5). As shown in our simulations, WaveCanCoh successfully detects sharp transitions in coherence that Fourier-based methods fail to resolve (e.g., Figure 2 right panel).
>
> 4. Theoretical Innovation and Practical Utility.
>
> Our work rigorously defines localized canonical coherence within the MvLSW framework (Definition 1, equation (8)) and provides consistent estimation and inference procedures (Algorithm 1; Theorem in Appendix B). The approach further extends naturally to causal analysis via lag-specific coherence (Causal-WaveCanCoh), allowing exploration of directed inter-group associations. This extension is also novel and absent in existing CCA or wavelet coherence literature.
>
> 5. Applicability to high-dimensional, real-world data.
>
> Finally, our application to LFP data showcases the ability of WaveCanCoh to reveal interpretable and behaviorally relevant coherence patterns across trials, scales, and time (Figure 4–5). Notably, the method identifies distinct coherence structures between correct and incorrect decisions that are missed by traditional methods using moving windows or global spectra.
>
> In summary, WaveCanCoh represents a novel and substantial advancement over existing methods by:
>
> (1) Extending canonical coherence to the nonstationary, multivariate, time-frequency domain; (2) Providing localized, scale-specific inference within the rigorous MvLSW framework; (3) Enabling causal (lag-specific) coherence analysis; (4) Demonstrating both theoretical soundness and empirical superiority in detecting dynamic group-level dependencies.
>
> We hope this clarifies the unique contributions of our work. In the light of your comments, we will improve our literature review in the revised version to highlight the novelties of our work.
>
> ### W2. Large-scale experiments with real data
> Thank you for the thoughtful suggestion regarding the need for larger-scale experiments and additional real-world validations. We fully agree that broader empirical evaluation can further strengthen the practical impact of our proposed method. Below, we would like to respectfully reclarify the experiment setting in our work.
>
> Our experimental design was guided by the goal of providing a comprehensive yet focused validation of WaveCanCoh from both theoretical and practical standpoints. The simulation section includes two carefully constructed scenarios: one that adheres to the assumptions of the multivariate locally stationary wavelet (MvLSW) framework and another based on a more general class of nonstationary AR(2) mixtures without any wavelet structure. This dual setup demonstrates not only that the method performs as expected under ideal conditions, but also that it remains robust when the assumptions are not adhered to, which is critical for assessing real-world applicability.
>
> On the real data side, we applied WaveCanCoh to high dimensional hippocampal LFP recordings during a memory guided behavior task. This dataset was chosen because it presents a prototypical example of multivariate nonstationary brain signals with complex temporal dynamics and behavioral variation. The analysis includes multiple channels grouped into anatomical subregions, comparisons across experimental conditions (correct vs. incorrect decisions), and multiscale frequency-resolved coherence patterns. As shown in Figures 4–5, the method was able to extract interpretable, scale-specific coherence signatures linked to cognitive performance, which would be difficult to detect using traditional approaches.
>
> We fully acknowledge the value of validating our method across a broader range of real-world datasets and plan to extend our work in this direction. Nevertheless, we believe that the current experimental results already provide strong evidence for the effectiveness, robustness, and interpretability of the proposed framework across both synthetic and real data settings.
>
> We hope this clarifies our rationale and appreciate the opportunity to discuss potential extensions.
>
> ***Thank you again*** for your valuable feedback. We look forward to continuing the discussion with you in the next stage, and please do not hesitate to let us know if you have any further concerns.

---

> > ### Author Response · Authors · 2025-08-05
> >
> > Dear reviewer,
> >
> > We hope our previous responses have sufficiently clarified the concerns you raised. If there are any remaining questions or suggestions, please do not hesitate to let us know — we would be more than happy to provide further clarification. We sincerely thank you once again for your time and effort in reviewing our work.

---

### Official Review · Reviewer_Jb5K · 2025-07-02

**Clarity:** 3
**Significance:** 3
**Originality:** 3
**Rating:** 4
**Confidence:** 4

**Summary:**

This paper introduces scale-specific wavelet canonical coherence (WaveCanCoh), a new framework to measure time-varying coherence between two clusters of multivariate time series. The authors target the problem of analyzing dynamic dependencies between groups of signals in a nonstationary setting. The main contributions of this paper include defining this new measure of coherence, developing a theoretically grounded algorithm for its estimation, and demonstrating its utility on simulated data. Overall, this paper will have broad potential in certain specific time series applications, especially in non-stationary time series data such as EEG. Since the Fourier transform is widely used on some time series, the wavelet-based approach proposed by the author is interesting.

**Questions:**

Does the effect of WaveCanCoh depend on frequency band design? For example, does the high-frequency part lead to low correlation?

**Ethical Concerns:**

["NO or VERY MINOR ethics concerns only"]

**Final Justification:**

The authors have addressed my questions. But I think my original rating is a fair grading.

**Limitations:**

The authors have addressed potential limitations.

**Quality:**

3

**Strengths And Weaknesses:**

Strength

1/ The paper is generally well-written and organized. The paper thoughtfully provides a detailed introduction to the technical background used, making this paper highly readable.

2/ The use of wavelet models to calculate the correlation of time series is relatively rare in ML research. Therefore, this work may provide new insights and vitality to time series researchers in the ML community.

3/ WaveCanCoh has strong innovation, extending Canonical Coherence to both time and frequency domains for the first time. This enables the model to extract non-stationary dependencies from some EEG signals.

4/ The paper’s technical quality is excellent. It provides a rigorous theoretical development and backs it up with thorough experiments.

Weakness

1/ I personally is not entirely sure how relevant this topic is to the NeurIPS audience. But I am open to be convinced otherwise.

2/ The manuscript includes a large amount of notation in the paper, which may not be friendly for the researchers to follow.

3/ Please add a citation where LSP first appears in the paper, or at least let readers know that the Fourier-based approach you mentioned above is LSP.

---

> ### Author Rebuttal · Authors · 2025-07-30
>
> Thank you very much for your detailed and professional review. Your comments have been extremely helpful in improving the quality of our paper. We also sincerely appreciate your recognition of the novelty of our work and its potential to inspire research in the machine learning community. Below, please allow us to provide some clarifications and responses to the concerns you have raised.
>
> ## Responses to the weakness
>
> ### W1.  Relevance to the NeurIPS program and audience
> As you rightly pointed out, the proposed method is effective in capturing the dependence structure between nonstationary signals. Nonstationarity is a typical characteristic of many types of multi-modal brain signals. Therefore, although the current paper only focuses on LFP data, our method can be directly applied to a wide range of neuroscience related studies, such as brain connectivity and brain-computer interfaces, topics that are central to the NeurIPS community.
>
> While we understand that much of the work at NeurIPS is closely tied to deep learning methodologies, we would like to emphasize that the existence of a theoretically justified statistical framework for capturing the dependence structure among multivariate time series is crucial for the success of many deep learning-based approaches. Our method is particularly well-suited for detecting time-varying dependence structures, which are often present in real world data and yet current methodologies are often unable to capture this dynamic aspect.
>
> As such, we believe our work has the potential to inspire and support the development of deep learning models for tasks such as time series classification, anomaly detection, and forecasting. Based on these reasons, we believe that this work aligns well with the scope and interests of the NeurIPS community in several important ways.
>
> ### W2.  Notations in the paper
> Thank you for pointing out this issue, you are absolutely right in pointing out that the notation is quite heavy. We have carefully reflected on your point, and while we believe that when we first introduce these quantities we need to preserve all dependencies (such as on the rescaled time $u$ and scale $j$) for mathematical clarity, however we agree that we can subsequently drop some of this cumbersome notation upon clear explanations. We will therefore review the notation used throughout the paper to ensure that each term is clearly defined and properly explained, in order to further improve the readability of the manuscript.
>
> ### W3.  About LSP method in the paper
> Thank you for raising this important point. We acknowledge that due to our efforts to stick within the mandatory page limit, the presentation of the LSP method in the manuscript has been very succinct and relegated to the Appendix C.2, which may have rightly caused some confusion, hence we appreciate the opportunity to clarify here.
>
> To the best of our knowledge, although a natural starting point as you have also observed, the LSP-based method used as a benchmark in our paper has not been previously formally developed in the literature for measuring time-varying canonical coherence between two sets of multivariate time series. This approach was specifically designed by us as a natural extension of classical Fourier-based canonical coherence methods, which assume stationarity and yield a global frequency-domain measure that discards temporal information. The absence of a method to track both time and frequency-varying canonical dependence motivated our development of a new, theoretically grounded framework.
>
> Our proposed WaveCanCoh addresses this gap by enabling the estimation of canonical coherence at each individual time point and across wavelet (i.e., frequency) scales. This dual localization contrasts with classical canonical correlation analysis, which provides a single global measure in the time domain, and Fourier-based coherence, which captures only frequency-domain information without temporal resolution.
>
> To provide a baseline for comparison, we introduce this Fourier-based Locally Stationary Process (LSP) method, which approximates time-varying canonical coherence by segmenting the time series into overlapping blocks and applying smoothing across the estimated canonical coherence values in the frequency domain. We will revise the manuscript to explicitly state that the LSP method is proposed in this work and clarify its role and novelty when it is first introduced.
>
> ## Questions
>
> ### Q1. Does the effect of WaveCanCoh depend on frequency band design? For example, does the high-frequency part lead to low correlation?
> Thank you for this insightful question. We would like to clarify that the effectiveness of WaveCanCoh does not inherently depend on the choice of frequency bands. Instead, the correspondence between wavelet scale $j$ and the actual frequency bands is determined by the sampling rate of the data, as illustrated in Figure 7 of the paper. Once the sampling rate is fixed, the relationship between scale and frequency is uniquely determined and remains unchanged—unless the data is explicitly resampled, which we also discuss as a minor limitation in Appendix D.3 (lines 497–500).
>
> Regarding the correlation in high or low frequency bands, WaveCanCoh does not assume stronger or weaker coherence at any particular frequency range. Rather, it is designed to reveal the scale-specific (i.e., frequency-specific) dependence structure that truly exists in the data. Whether stronger coherence appears in high-frequency or low-frequency components entirely depends on the underlying dynamics of the signals being studied.
>
> For example, although outside of the NeurIPS scope but still relevant for our methodology, in the context of financial time series, if two markets react rapidly to global news, their hourly price movements may show high coherence—this would be captured in the high-frequency bands. On the other hand, if two assets exhibit similar long-term trends, WaveCanCoh will detect stronger coherence in the low-frequency bands. Similarly, in neuroscience applications, certain types of brain activity or connectivity may manifest predominantly in either the theta or gamma bands, and WaveCanCoh provides a natural way to analyze such dependencies over both time and scale.
>
> In summary, WaveCanCoh is data-adaptive: it does not bias the narrative towards any particular frequency range, but instead captures whichever scale-specific temporal dependence is present in the signals. This makes it a powerful tool for analyzing a wide range of dynamic multivariate systems.
>
> **Once again**, we sincerely thank you for your positive evaluation of our work. We hope that our responses have helped to clarify the manuscript, and we would be very happy to engage in further discussions with you in the next stage.

---

> > ### Comment · Reviewer_Jb5K · 2025-08-04
> >
> > Thanks the authors for answering my questions and clarifying my initial doubts in details. I have no further queries.

---

> > > ### Author Response · Authors · 2025-08-05
> > >
> > > We sincerely thank you again for your valuable comments and suggestions. We truly appreciate the time and effort you dedicated to reviewing our work. Please do not hesitate to let us know if you have any further questions or concerns—we would be more than happy to provide additional clarifications.

---

### Official Review · Reviewer_HjCP · 2025-07-02

**Clarity:** 3
**Significance:** 4
**Originality:** 4
**Rating:** 5
**Confidence:** 4

**Summary:**

In this work, the authors describe a new framework which uses wavelets to measure canonical coherence between sets of multivariate time series. Their approach, called wavelet canonical coherence (WaveCanCoh) builds from the local wavelet spectral (LWS) matrix along with eigenvector analysis typically seen in canonical correlation analysis (CCA) in order to quantify associations given in terms of spectral associations, rather than standard temporal associations.  The authors demonstrate their method on simulated time series and real local field potential (LFP) data.

**Questions:**

1. CCA has seen quite a few applications in fusion across imaging modalities (e.g. fMRI+EEG). Could the proposed method be used for multi-modal fusion of time-series data with different sampling rates? I fear the scale-specifity of the method might limit this potential?
2. How does the provided estimation procedure scale in terms of time and memory complexity?  How does that compare to standard CCA? Eigenvalue decompositions in particular are notoriously intensive both in time and memory.
3. How do the asymptotic guarantees given for the WaveCanCoh estimator apply in practice? Time time points and smoothing needing to approach infinity doesn’t provide a good idea of the practical upshot of longer sequences or increased smoothing.

**Ethical Concerns:**

["NO or VERY MINOR ethics concerns only"]

**Final Justification:**

I would recommend this work for acceptance. I was initially quite happy with the quality of this work, and all of my concerns have been adequately addressed. The author responses have been more than sufficient, and I would recommend that this work be accepted. I do agree with some of the negative reviews that larger-scale experiments would improve the quality of the paper; however, I do believe the quality of the experiments and other work is sufficient and not a significant detractor from the overall impact of the work.

**Limitations:**

As stated earlier, I cannot find any discussion of the limitations in this work except for one sentence in the appendix. My questions regarding computational complexity and scale-specificity hint at some possible limitations that may arise, but a more thorough discussion of limitations in the main body of the text is required.

**Paper Formatting Concerns:**

Some of the images are small and captions are difficult to read; however, I have no other issues regarding format concerns.

**Quality:**

3

**Strengths And Weaknesses:**

# Quality
Overall, the quality of this work is good. The work is technically sound, rigorous, and correct as far as I have found. Limitations are not provided, which lowers this to a good for me, but I am willing to change this if I missed them or they are included in a revision.
## Strengths
To the best of my knowledge, the theoretical framework and its associated proofs are all technically sound. The provided experiments are rigorous and demonstrate the benefits of the proposed method.
## Weaknesses
Although the authors claim to discuss limitations, I could not find the referenced material anywhere in the text or appendix.

Additionally, the related work section is limited in its discussion of actual related literature. Although the authors are right and I know of no exactly analogous methods which do CCA as such with Wavelets, there are plenty of techniques that use the frequency domain or wavelet-based transformations to study associations between time-series, especially in neuroimaging e.g. [1,2,3,4]. These methods (and others) are indeed related—at least in spirit—and ought to be mentioned. I think using the related work section to introduce the theoretical background is an unconventional choice.

[1] Skidmore, F., et al. "Connectivity brain networks based on wavelet correlation analysis in Parkinson fMRI data." Neuroscience letters 499.1 (2011): 47-51.

[2] Ghuman, Avniel Singh, Jonathan R. McDaniel, and Alex Martin. "A wavelet-based method for measuring the oscillatory dynamics of resting-state functional connectivity in MEG." Neuroimage 56.1 (2011): 69-77.

[3] Sato, Joao Ricardo, et al. "A method to produce evolving functional connectivity maps during the course of an fMRI experiment using wavelet-based time-varying Granger causality." Neuroimage 31.1 (2006): 187-196.

[4] Tan, Qitao, et al. "Frequency‐specific functional connectivity revealed by wavelet‐based coherence analysis in elderly subjects with cerebral infarction using NIRS method." Medical physics 42.9 (2015): 5391-5403.

# Clarity
Overall, the clarity of the work is good. I am willing to raise this score if my concerns are addressed.
## Strengths
Overall, this work is well-structured and well-written. I have no issues with the structure of the paper or the writing quality.
## Weaknesses
Many of the figures have been scaled down to fit the space. While this is fine, much of the text in the figures (especially in the legends) is quite small and difficult to read when printed. For example:
* The text of the legend in figures 2 and 3 is especially hard to read.
* Figure 4 is clearer, but larger text or scaling up the figure would be an improvement.
* Figure 10 in the appendix also has quite small text, especially “Odor presentation”.
# Significance
Overall, the significance of this work is excellent. Good work!
## Strengths
I believe this method holds great promise for applications in M/EEG and other neuroimaging analyses. Upon reading this paper, I immediately want to try to compare this method to other metrics for studying associations in EEG/MEG, such as wPLI. I am especially excited to see if this approach could be used for multi-modal fusion of time-series from different acquisitions.
## Weaknesses
I have no weaknesses to discuss in terms of significance.
# Originality
Overall, I would call the originality of this work fair to good (I am going to put fair on my score, but I am leaning toward good). I am willing to improve this score if the authors ground their work in the literature and provide a convincing argument for why this method stands out from other wavelet-based connectivity approaches.
## Strengths
As far as I have found, this work is original in terms of performing CCA to find multivariate time-series associations via wavelets.
## Weaknesses
Despite the apparent originality, the authors have neglected to ground the approach in other wavelet-based measures for brain connectivity. This work would be significantly improved by a more thorough grounding in the literature.

---

> ### Author Rebuttal · Authors · 2025-07-30
>
> Thank you for your detailed and constructive review. We truly appreciate your deep understanding of the field, and many of your comments have been very insightful for us. We are grateful for your recognition of the novelty and practical relevance of our work. Our intention was to develop a framework that is both simple and novel, yet highly useful in practice. In the following, please allow us to respond and clarify the concerns you have raised.
>
> ## Responses to the weakness
> ### W1. Discussion on the limitations
> We stated the limitation of our approach in Appendix D.3 (lines 497–500) as you pointed out, although we acknowledge that due to our concern for the strict space constraints, the discussion was very succinct leading to a lack of clarity for which we apologise. We will revise this part and include a dedicated paragraph on method limitations in the main text. Below, we outline the discussion we plan to further incorporate.
>
> While our WaveCanCoh framework provides a robust, nonparametric approach for quantifying scale-specific time-varying canonical coherence between two sets of nonstationary multivariate time series, it is inherently limited to a fixed set of wavelet scales. This restricts its flexibility in applications requiring precise frequency localization or alignment with arbitrary bands. As shown in Figure 7, the mapping between scales and true frequency depends on the sampling rate and signal spectrum. For signals with broad frequency content or low sampling rates, certain bands may be poorly resolved. One practical solution is downsampling when a high sampling rate is available, allowing better alignment between scales and target frequency bands. However, due to time-frequency trade-offs in wavelet analysis, fully flexible frequency resolution remains challenging. Future work may consider adaptive or overcomplete wavelets to improve frequency targeting while retaining nonstationary modeling capabilities.
>
> ### W2. Related literature review
> We agree with your suggestion that more background on previous time series studies using wavelets, as well as a clearer articulation of the importance and novelty of our contribution, would strengthen the paper. In the revised version, we will provide a more comprehensive and structured review of relevant literature to better position our work in the existing body of research. To briefly clarify the key novelty of our work: while prior methods such as wavelet coherence focus on dependencies between pairs of univariate signals, our framework—Wavelet Canonical Coherence (WaveCanCoh)—is the first to define and estimate canonical coherence between two sets of nonstationary multivariate time series in a  ***joint*** time- and scale-localized manner. This enables us to capture interpretable, global inter-group interactions and their dynamics, rather than just localized pairwise associations. Our approach extends canonical coherence analysis into the wavelet domain, providing a principled framework for nonstationary dependence modeling at the group level—something that existing methods, to our knowledge, do not address. As you suggested, we will discuss a more complete set of references to related studies in the revised version of the paper to better place our contribution within the existing literature.
>
> ### W3. Label visibility for some figures
> We acknowledge that the font size in several of the figures you mentioned is a bit small and may be difficult to read especially when relying on a printed version. We sincerely apologize for this and fully commit to increasing the font size in the revised version to improve legibility. We will also ensure that the quality of all figures throughout the paper meets a consistent and readable standard.
>
> ## Questions
> ### Q1. Application to multi-modal fusion
> Due to its flexibility, this framework can be easily applied on the signals obtained from other modalities, e.g., fMRI, MEG and EEG. In fact, we have already implemented this method on EEG data from subjects with and without ADHD data, and obtained neurologically promising results. In this paper, we chose to use the LFP data because it has typical nonstationary features, which allow us to illustrate both the motivation and novelty of our proposed approach. Although we have not experimented with it ourselves, we believe the method can also be used for multi-modal fusion of time series data with different sampling rates, as these will be appropriately captured by the spectral representation and subsequently formed quantities.
>
> Specifically, when applied to multi-modal data, the WaveCanCoh framework can be adapted to analyze cross-modal interactions by treating each modality as a component in the multivariate signal sets. Suppose we have $\mathbf{X}_t = (X^{(1)}_t, \ldots, X^{(P)}_t)^\top$ denoting signals from modality 1 (e.g., EEG) and $\mathbf{Y}_t = (Y^{(1)}_t, \ldots, Y^{(Q)}_t)^\top$ denoting signals from modality 2 (e.g., MEG or fMRI). Scale-specificity would not be a problem here, but we should be careful to match the scales and frequency bands when the aim is to draw conclusions in the Fourier frequency domain. For example, if the sampling rate of $\mathbf{X}_t$ is $20Hz$ and that of $\mathbf{Y}_t$ is $10Hz$, then WaveCanCoh between $\mathbf{X}_t$ and $\mathbf{Y}_t$ at scale $j=1$, $\boldsymbol{\rho}_j (t)$ represents the time dependent coherence between the components of $\mathbf{X}_t$ at $5-10Hz$, and the components of $\mathbf{Y}_t$ at $2.5-5Hz$. Furthermore, this may also provide a way to identify how fluctuations in longer-term dynamics (low frequency components) of signal from modality 1 may have an impact on the amplitude of the shorter-term dynamics (high frequency components) of modality 2. The canonical directions obtained through WaveCanCoh estimation will therefore highlight maximally correlated linear combinations of the channels from each modality, thus allowing the framework to capture joint dynamics that may not be apparent in direct pairwise analysis. In summary, we believe multi-modal fusion of time series data with WaveCanCoh is feasible, easy to implement and may provide subtle information on joint dynamics.
>
> ### Q2. Computational complexity of the method
> The estimation of spectrum step leads to a time complexity $\mathcal{O}(JT(P+Q)^2)$ (where the number of scales $J$ is used most commonly as $J=\log_2(T)$). The eigenvalue decomposition computation step takes the total time complexity to $\mathcal{O}(JTd^3 )$ (let $P \approx Q =d$), to be compared to that of the standard CCA,  $\mathcal{O}(T d^2 + d^3)$. The proposed algorithm is naturally more expensive than standard CCA, since it is designed to obtain time-localized and scale-specific results. In our practical experience with WaveCanCoh, the spectral estimation step is extremely fast, with virtually all of the computational time being spent on the eigenvalue decomposition step. Although compared to standard CCA, our method produces a set of results at each time point, resulting in increased computational burden, in our experiments with $T = 1024$, a single replicate is completed within 2.5 seconds on a standard personal computer (Apple Mac, 16GB RAM, 6-core CPU) without resourcing to parallel computing or to use a cluster. Therefore, we believe the computational expense would not be a major concern, even in large-scale experiments.
>
> Moreover, computations for WaveCanCoh can be streamlined by storing canonical vectors and coherence values at each time and scale, with the requirement being $\mathcal{O}(JTd^2)$, and the memory complexity $\mathcal{O}(d^2)$ for standard, one time point global estimate.
>
> ### Q3. Asymptotic guarantees for the WaveCanCoh estimator
> Firstly, we would like to clarify that the conditions assuming $M, T \to \infty$ with $M/T \to 0$ are theoretical requirements to rigorously guarantee the consistency of the estimator towards the true, unknown quantity. In practice, a commonly recommended choice for the smoothing parameter is $M = \lfloor \sqrt{T} \rfloor$, where $\lfloor \cdot \rfloor$ denotes the floor function. For example, in our experiments with $T = 1024$, we used $M = 32$, a value is that is in line with those reported in the related wavelet process literature.
>
> We understand your could concern regarding the convergence of the estimator for low temporal resolution data (e.g., fMRI). As a response, we would like to highlight the results shown in Figure 2 (left), where the convergence of the estimator was empirically verified. In our view, both $T = 1024$ and $M = 32$ are moderate in size, and the estimator performs reliably under these conditions. Moreover, based on our prior experience with this type of wavelet spectral estimation, we have observed stable convergence even with much (8-fold) smaller values of $T$ and $M$. Thus, based on both theoretical and practical arguments, the WaveCanCoh estimator remains practically effective unless the time series is extremely short (e.g., $T = 20, 30$).
>
> If further improvements in convergence are desired for shorter time series, a practical strategy is to apply periodic extension to the original time series to estimate the local spectral matrix, and then to perform the eigenvalue decomposition step using only the original time segment. Since the computational cost of spectral estimation is low, this approach offers a feasible way to improve performance without incurring significant computational overhead.
>
> ***Once again***, we sincerely thank you for your insightful comments and the time you dedicated to reviewing our work. We hope that our responses above have fully addressed your concerns. If you have any further questions or suggestions, please do not hesitate to let us know—we would be more than happy to continue the discussion in the next stage.

---

> > ### Comment · Reviewer_HjCP · 2025-08-05
> > **Thank You - More Concrete Feedback would be Appreciated**
> >
> > Hello again. Thank you very much for your detailed response to my comments. I would like to follow-up to each of your responses individually.
> >
> > ## W1. Discussion on the Limitations
> >
> > I am very happy to see this extended discussion of limitations, and glad it will be incorporated into the main text. I think the paragraph you have provided is great; however, I would ask, that you perhaps mention at least one real example, if any, with "broad frequency content or low sampling rates". Functional MRI with a low TR might be one obvious choice here for at least the latter condition.
> >
> > ## W2. Related literature review
> >
> > I am glad that you agree about needing to ground the work more substantially in the literature. Could you please provide a brief summary of which citations you will be using in this discussion and how they relate to and contrast with your work?
> >
> > Thank you for the clarification regarding the novelty of this work as well! I think adding a more substantial discussion will help other readers appreciate the novelty of your work in relation to other existing methods.
> >
> > Per my review, I am happy to increase my evaluation from "Fair" to "Good" and will increase to "Excellent" once I get a more concrete idea of which citations you are using for related work discussion and how they relate to and contrast from your work.
> >
> > ## W3. Label visibility
> >
> > Thank you for making this adjustment. Since Neurips is not allowing images in responses, I appreciate that we will have to take your word at this stage. To the area chair, please see this example as to why updated manuscripts and images should be allowed in the discussion phase.
> >
> > ## Q1. Application to Multi-Modal Fusion
> > Just for my sanity, when you say _"we have already implemented this method on EEG data from subjects with and without ADHD data"_ I'm a bit confused about what the modalities are here? EEG and LFP? EEG and Longitudinal clinical measures of ADHD?
> >
> > The rest of your response is great - thank you very much. I think the potential for multi-modal fusion is a particularly exciting implication of this work, though certainly the need for matching in order to make inferences in the frequency domain may introduce some wrinkles say in EEG/fMRI fusion.
> >
> > ## Q2. Computational Complexity
> >
> > Thank you for providing this analysis - I am pleased with your discussion here and if you agree, I think adding a small section to your appendix containing this material will improve the quality of this work.
> >
> > ## Q3. Asymptotic Guarantees
> >
> > Thank you for your response! Indeed I was concerned about the practical upshot for methods with a relatively low sampling rate such as fMRI. I am happy with your response here, and have no further questions on this front.

---

> > > ### Author Response · Authors · 2025-08-06
> > >
> > > We sincerely appreciate your thoughtful feedback. We're glad to continue the discussion and provide additional details addressing the concerns you highlighted.
> > > ## W1. About Limitation
> > > We are pleased that you found the discussion in this section valuable, and we truly appreciate your thoughtful suggestion. We fully agree that providing a simple example would greatly enhance clarity and help readers grasp the idea more intuitively. In line with your suggestion we will make sure to include it.
> > > ## W2. Related Literatures
> > > We fully agree that the absence of a more detailed comparison with existing methods in the initial version may have prevented the strong innovations of our approach from being fully appreciated. In the revision, we will enhance the related work discussion in three key areas to better emphasize the novelty and practical value of our approach.
> > >
> > > Firstly, we will elaborate on the further inference that can be derived from WaveCanCoh versus the limitations of typically employed methods such as CCA, Fourier-based coherence, and existing CCA-based time series approaches. While classical CCA and Fourier coherence [Hotelling, 1936; Brillinger, 2001; Chang \& Glover, 2010] are widely used to study linear dependence and spectral connectivity, they assume stationarity and cannot capture localized, time-varying, and cross-scale interactions. To our knowledge, WaveCanCoh is the first to embed CCA into a wavelet framework for simultaneously analyzing localized multiscale associations between multivariate nonstationary time series. We will further clarify this point with additional references such as [Sun \& Bollt, 2014; Sakoglu et al., 2010].
> > >
> > > Secondly, we will expand the comparison with existing wavelet-based methods. Prior studies primarily focus on within-network coherence in multivariate systems or pairwise coherence between univariate channels [Lachaux et al., 1999; Bruns, 2004; Chang \& Glover, 2010]. Several wavelet-based connectivity approaches have been proposed for fMRI [Skidmore et al., 2011; Sato et al., 2006], MEG [Ghuman et al., 2011], and NIRS [Tan et al., 2015], but these typically rely on univariate or bivariate measures and lack a principled multivariate framework. In contrast, WaveCanCoh quantifies structured dependencies between two multivariate subsystems, making it well-suited for analyzing dynamic inter-regional interactions across multiple channels. This level of granularity is especially important in neuroscience, where the goal often involves inferring connectivity between functionally defined brain regions. To enhance this distinction, we will include additional references such as [Bullmore \& Sporns, 2009; Knyazeva et al., 2011; Usama et al., 2022; Liao et al., 2010].
> > >
> > > Finally, we will briefly discuss recent wavelet-based deep learning frameworks to highlight the broader potential of our method. Wavelet representations have been successfully incorporated into modern architectures such as WaveNet [van den Oord et al., 2016], WaveletNet [Jing et al., 2018], and FEDformer [Zhou et al., 2022] to enhance time-frequency modeling and scale-awareness. Additionally, wavelet decomposition has proven effective in neural models for EEG classification [Xin et al., 2022; Thuwajit et al., 2022], frequency-based emotion recognition [Zheng & Lu, 2015], and time series forecasting [Yu H. et al., 2024]. These works underscore the flexibility of wavelet-based modeling and suggest that WaveCanCoh could serve as a principled statistical foundation for future cross-domain or hybrid learning applications.
> > > ## W3. Label Visibility
> > > We sincerely appreciate your understanding and will ensure the final version reflects the proper adjustments we discussed.
> > > ## Q1. Multi-Modal Fusion
> > > We apologize for the confusion and appreciate the chance to clarify. Here, we intended to convey WaveCanCoh is easily adaptable to other modalities, and we have applied it to EEG data. However, you are correct that this remains a single-modality analysis using EEG from 30 children with ADHD and 20 healthy controls. “With and without ADHD” refers to diagnostic groups, not different modalities.
> > >
> > > While we have not yet applied the method to cross-modal settings (e.g., EEG vs. MEG), the framework is conceptually well-suited for such applications, as discussed in our earlier response, and many exciting applications remain to be explored.
> > > ## Q2. Computational Complexity
> > > We fully agree that clarifying the computational complexity is important and will include this in the appendix of the revised version. In fact, we believe that your other suggestions align well with the content we aim to communicate, and will help strengthen WaveCanCoh in both practical and theoretical aspects.
> > > ## Q3. Asymptotic Guarantees
> > > We are pleased that our explanation helped clarify this point.
> > >
> > > Many of your suggestions have greatly improved the clarity and quality of our paper, appreciate it! We're happy to continue the discussion if further clarification is needed.

---

> > > > ### Comment · Reviewer_HjCP · 2025-08-06
> > > > **Thank you!**
> > > >
> > > > Thank you very much for your feedback! I am pleased with the response of the authors here and will adjust my scores accordingly.

---

> > > > > ### Author Response · Authors · 2025-08-06
> > > > >
> > > > > Thank you so much for your encouraging feedback! We sincerely appreciate your thoughtful comments and the time you devoted to reviewing our manuscript, and we will carefully reflect on the points raised to further strengthen the paper.

---

### Official Review · Reviewer_RXsG · 2025-07-03

**Clarity:** 3
**Significance:** 2
**Originality:** 3
**Rating:** 5
**Confidence:** 4

**Summary:**

The authors present a method for quantifying multi-channel correlations at different time scales. The method produces scale-specific components for each channel via wavelet decomposition and then applies CCA to the resulting components (with some additional pre-processing). The method is presented in the context of canonical coherence for stationary processes and outlines both the model and estimation procedure. The model is then validated with simulated data using both wavelet-based and non-wavelet based nonstationary generative models and is then used on LFP data from mice performing a 2AF task at a pre-selected number of scales, and lags.

**Questions:**

- I would recommend changing the word “clusters” used throughout the manuscript to refer to distinct sets of channels since “clustering” is unrelated to the present paper.
- The a and b elements in the inline text under equation (8) are bolded but the authors previously used the convention of non-bold symbols for scalars earlier in th paper. The authors should maintain a single convention throughout the paper.
- Please explain A_{jl} in equation (14). It is not mentioned anywhere in the text
- What kind of wavelet was used for the analysis in section 6?
- Could the authors include the benchmark I mentioned in S&W?
- Could the authors independently validate the results with another approach and/or give a deeper explanation for what the results mean and why they demonstrate the validity or value of their method? Alternatively, could the authors demonstrate that another method could not be used? This is perhaps the most important point to address for me to increase my score.
- It is not stated if the authors controlled for multiple comparisons in their LFP data analysis. They should control for multiple comparisons in this case.

**Ethical Concerns:**

["NO or VERY MINOR ethics concerns only"]

**Final Justification:**

the authors have satisfied all of my concerns. As such I think an increase in score is warranted.

**Limitations:**

The authors didn’t list any limitations. They could provide ways of further improving on their method and suggest scenarios where their method would not work. They could also comment on the scalability of their method and the suitability of their method for exploratory or discovery-based analysis.

**Quality:**

2

**Strengths And Weaknesses:**

The paper is organized clearly and logically developed and is mostly well-written, with a few omissions that could be corrected rather easily. I would imagine the method being useful to some researchers but could require some considerable guidance on how to choose wavelets, scales, and averaging window sizes M. Using specific windows of analysis to deal with nonstationary data is a pretty intuitive step for many neuroscientists but developing an approach explicitly in a wavelet setting is certainly more rigorous. However, it is not entirely clear from the text what impact it could have, however. First, the analysis was not compared against the most obvious benchmark, which would have been to apply canonical coherence on a moving window and evaluate the SNR against their method. Second, the analysis of LFP data was extremely limited with ambiguous ROI. Specifically, the results were not independently validated or interpreted either physiologically or algorithmically.

---

> ### Author Rebuttal · Authors · 2025-07-31
>
> Thank you for your insightful review. We appreciate your time and believe that addressing your comments will greatly improve the clarity and quality of our work. We're also grateful for your recognition of WaveCanCoh's novelty and practical value. Below are our detailed responses.
>
> ## Responses to the weakness
> ### W1. Choice of wavelets, scales and kernel size, as well as their impact
> Thank you for highlighting this. We agree that a clearer explanation will better guide researchers applying the method.
>
> The choice of wavelets primarily depends on the characteristics of the signal, e.g., Haar wavelets are suitable for signals with sharp transitions, while Daubechies wavelets are suggestted for smoother signals. In practice, unless the signal shows very distinct features, results tend not to vary much across different wavelets. In our LFP analysis, we used Daubechies wavelets with 4 vanishing moments (Db-4), which offers a balanced choice for diverse features. To ensure robustness, we compared results using Haar and Db-8 for LFP data analysis. The outcomes were largely consistent, supporting the reliability of the findings obtained with Db-4.
>
> The choice of scale depends on the frequency range of interest. In our study, we focused on scales $j = 3$ to $j = 7$, as they cover the primary frequency bands where neural activity is most prominent. WaveCanCoh makes no assumptions on coherence strength at any scale but aims to capture true scale-specific (frequency) dependence driven by data,
>
> For window size $M$, the recommended value in wavelet literature is $\lfloor \sqrt{T} \rfloor$, which offers a good trade-off between bias and variance in spectral estimation. Large deviations from this may introduce bias and reduce estimation accuracy. We have verified this behaviour through extensive simulations for our method.
>
> ### W2. Comparison benchmark
> Thank you for this important point. In our study, we implemented a Fourier-based coherence benchmark using a moving window, which referred to in text under the acronym LSP (Locally Stationary Fourier Process), as described in Appendix C.2 in the paper. This benchmark was evaluated using simulated data generated from a mixture of AR(2) processes. We would like to emphasize the key differences between the LSP and WaveCanCoh. Specifically, in LSP, coherence at each time point is approximated using the surrounding window, making the results sensitive to e.g., window size and kernel choice. More importantly, due to the global nature of the Fourier basis, LSP struggles to capture localized or transient features—especially in the presence of sharp transitions, a point illustrated in Fig. 2 right panel.
>
> We agree that including this benchmark in LFP data analysis would strengthen the comparison. In the revised version, we have incorporated LSP into the analysis and produced results analogous to Fig. 4 and Table 1. As anticipated, the LSP approach fails to effectively capture the sharp trend change at the onset of odor presentation (much smoother). We are fully committed to updating the manuscript with the revised results.
>
> ### W3. Validation and interpretation of the results
> Very good concerns! We'd like to address them from both algorithmic and neuroscientific perspectives.
>
> Firstly, we clarify about the 'ROI'. The selection of the two brain regions was not outcome-driven, but based on prior knowledge: the hippocampus is critical for learning, memory, and decision-making. Despite extensive study, how its neural activity differs between correct and incorrect decisions remains unclear. This dataset offers millisecond-level resolution in neural and behavioral events under a decision task, making it well-suited to explore this. We have revised Fig. 1 to include a panel showing the ROI and electrode locations, clarifying the recording sites. The two selected regions are spatially distant, our aim is to examine whether they communicate, and whether this communication differs by behavioral outcome.
>
> WaveCanCoh was developed to get detailed insights into dependence structure across time and frequency, enabling accurate detection of neural activity differences across behaviors (correct vs. incorrect). As shown in Table 1, following odor presentation ($t = 0$), clear differences emerge between behaviours at specific time points and frequency bands, while no significant differences are seen before $t = 0$, as expected.
>
> Our interpretation is that if significant differences were observed across all frequency bands after $t = 0$, the use of a frequency-domain (wavelet) method would not be particularly necessary. Similarly, if significant differences appeared at all time points within a specific frequency band, the value of a method with fine temporal localization would be diminished (which means classic Fourier based method can do this). The results demonstrate that it is necessary to have a method like WaveCanCoh that can capture detailed information, specifically, which frequency components differ and when they differ between behavioral conditions. We believe this capability is essential for advancing our understanding of the neural mechanisms underlying behavior and task performance. For example, as shown in Table 1, at scale $j = 5$ and $t=0.5$, the canonical coherence between the selected brain regions is significantly higher in correct trials compared to incorrect trials. This suggests that weaker communication between these regions may contribute to incorrect decisions. A reasonable interpretation is that the interaction between these regions may play a crucial role in decision-related processes involved in the task.
>
> To further validate our findings and assess the reliability of the algorithm, we conducted a permutation test on simulated data. Two datasets with different spectral structures were generated: one matching Simulation 1 (Figure 2, left) and another with a slight modification ($c(u) = 2$ for $u \in (0, 1)$; see Appendix C.1), introducing coherence differences in the first half of the time period while keeping the second half identical. We applied the permutation test (Algorithm 2) at several time points to test $H_0$: no difference in canonical coherence, versus $H_1$: significant difference. Based on 1000 replicates, the test yielded a Type I error rate of 2.1\% and a Type II error rate of 1.8\% (i.e., 98.2\% power), confirming its validity and supporting the reliability of the real-data results. Moreover, we also applied  the permutation test on proposed Fourier-based benchmark . We found that it was mostly unable to detect differences between correct and incorrect trials. This suggests that the low temporal resolution inherent in the moving-window approach may obscure meaningful temporal dynamics, potentially leading to incorrect interpretations of the underlying neural mechanisms.
>
> Finally, from a neuroscience perspective, the results are consistent with prior research using the same dataset in which correct and incorrect decisions could be differentiated using the spiking activity data (Allen et al., 2016; Shahbaba et al., 2022). We extend prior results by achieving similar differentiation using LFP data alone—a noisier and more challenging signal to classify. Notably, the distinction between trial types discovered by the model within the 15.625 - 31.25 Hz frequency range is in line with previous work showing learning-related signals in a similar band (Gattas et al., 2022). Further, a key innovation of the present method is that it can take the spatial (anatomical) location of LFP electrodes into account. This advance can provide novel insights into the specific contributions of sub-networks within a brain region or circuit, as demonstrated here for the hippocampus, which can then be mapped onto known characteristics of the network such as anatomical gradients.
>
> ### W4. Limitations of the proposed method
> Please refer to W1. of responses to the reviewer HjCP, thanks for your understanding.
>
> ## Question
> ### Q1. The use of the term 'clusters’
> Thank you for your suggestion, we will revise this to 'sets' to make it clearer.
> ### Q2. Bold symbol inconsistency
> Thanks for pointing this out, it will be fixed.
> ### Q3. Interpretation for $A_{jl}$
> Sorry for the lack of further information. The wavelet periodogram provides an asymptotically unbiased estimator for a linear combination of the true spectral quantities across scales. The matrix ${A}$, whose $(j,l)$ entry is $A_{jl}$, encodes the bias correction needed to recover a well-behaved estimate of the true spectra at scale $j$ from the set of smoothed raw periodograms across scales $\{\tilde{\mathbf{I}}_{l, k}\}$, at each time point ($k/T$). This leads to an asymptotically unbiased spectral estimator as given in equation (14).
>
> Specifically, $A_{j l}=<\Psi_j, \Psi_l>=\sum_\tau \Psi_j(\tau) \Psi_l(\tau)$ for $j, l \in \mathbb{N}$ where $\Psi_j(\tau) = \sum_k \psi_{j, k} \psi_{j, k-\tau}$ is the discrete autocorrelation wavelet.
>
> ### Q4. Wavelet used for Section 6
> Sorry for this omission. The wavelet used here is the Db-4,  which is a robust choice for extracting varying signal features (refer to W1).
>
> ### Q5. Multiple comparisons in data analysis
> We fully agree that accounting for multiple comparisons is an important aspect, with one being potentially interested in more or less conservative testing, as dictated by the context and analysis aims. Our permutation test results in Table 1 report the p-values, and the significance ($^\star$) follows standard convention. In the light of your suggestion, we will also comment on the results associated to having applied a Bonferroni correction, with the number of tests determined by the 5 scales and 4 time points of interest. We note that with one exception ($t^\star=1.0$ at scale 6), all other significant differences still hold under this conservative regime.
>
> ***Thank you again*** for your thoughtful feedback. We hope our responses provide clarity and welcome further discussion.

---

> > ### Comment · Reviewer_RXsG · 2025-08-04
> >
> > Thank you to the authors for trying to be thorough with their responses. I think there may be some miscommunication about what I’m looking for, however.
> > To clarify, by “ROI” I meant Return on Investment”, not “region of interest” I apologize for the confusion. My main point is that it was not obvious from the text whether it’s worth the trouble to use such a sensitive analysis technique. For example, most neuroscientists will tune the window size to match the temporal resolution they need.
> > > Permutation test on simulated data: I’m honestly not sure how it helps your case to run statistics on an obvious result that you constructed yourself. I was already quite convinced of this result from Figure 2. What I didn’t see, and what helps somewhat, is the stats you ran on the Fourier method for the LFP data. Please provide more details though as it would be awesome to see that you made a good-faith effort to implement a standard analysis technique that fails where yours does not.
> > >I’m not sure if it makes sense to call the LFP data “noisier”. That simply isn’t true in every case. In fact there are well documented cases of decodability of LFP being greater than that of spikes.
> > > It’s also not clear from your method how it can “take the spatial location of LFP electrodes into account”, not how this is a “key innovation of the present method”. None of this was explicitly mentioned in the paper. I see a spatial information in Figures 1 and 5, but I don’t see anything that this has anything to do with your method, but rather to do with the data. I also don’t see how you have taken this information into account. Please clarify.

---

> > > ### Author Response · Authors · 2025-08-05
> > >
> > > We sincerely appreciate your follow-up questions and the opportunity to further clarify your concern. We'd like to provide clarification from four key aspects.
> > >
> > > ## About the ROI
> > > We apologize for the misunderstanding and we now understand that your primary concern is related to the necessity of the proposed method. While tuning the window size in Fourier-based methods can offer some improvement in temporal resolution, all points within the kernel are still smoothed approximations. We believe you would agree that using extremely short windows to capture sharp transitions is neither statistically stable nor meaningful in terms of frequency resolution, albeit we agree with you that this is an often adopted working solution. More importantly, since the same window is applied across all frequencies, the method cannot adapt to frequency-specific features. In contrast, wavelet methods naturally adjust the window size across scales, offering more precise time-frequency localization. Very recent literature advocates with practical examples that these features make wavelet-based methods particularly well-suited for nonstationary signals such as neural data, where crucially frequency content also varies over time.
> > >
> > > ## Fourier method on LFP data
> > > Due to space limitations in the initial response, we were unable to provide further details on the additional experimental results. Pls allow us to expand on them below. We use a window size $M=50$ points with moving step $d = 1$ to approximate the coherence across entire trial ($T=4000$). We focus on the frequency bands corresponding to those in WaveCanCoh analysis. We generate the results corresponding to Figure 4 and Table 1 (having conducted the same permutation test).  We observed a similar behaviour to that in Figure 2 (right): the Fourier-based method was unable to capture the abrupt change in coherence at stimulus onset, resulting in overly smoothed estimates. Permutation test results further suggest that this approach lacks the sensitivity to distinguish between correct and incorrect trials (pls see below). The likely explanation is that the smoothed approximation may have masked the true differences between correct and incorrect trials.
> > > | j (Freq Band)       | -1.0           | -0.5           | 0.5            | 1.0            |
> > > |---------------------|----------------|----------------|----------------|----------------|
> > > | 3 (62.5–125Hz)      | -0.065 (0.886) | -0.002 (0.901) | -0.009 (0.991) | -0.060 (0.471) |
> > > | 4 (31.25–62.5Hz)    | 0.001 (0.128)  | -0.044 (0.141) | 0.010 (0.056*) | 0.010 (0.991)  |
> > > | 5 (15.63–31.25Hz)   | 0.009 (0.470)  | 0.003 (0.970)  | 0.008 (0.999)  | 0.003 (0.983)  |
> > > | 6 (7.81–15.63Hz)    | -0.001 (0.512) | -0.002 (0.052*)| 0.002 (0.901)  | 0.000 (0.842)  |
> > > | 7 (<7.81Hz)         | 0.001 (0.094)  | 0.001 (0.121)  | -0.004 (0.754) | -0.002 (0.901) |
> > >
> > > ## Properties of LFP data
> > > Thank you for your comment. We agree that the relative noisiness of LFP versus spiking data can vary depending on experimental context and brain region. Our intention was not to make a general claim, but to highlight that in this particular setting—hippocampal recordings during a cognitively demanding non-spatial task—LFP signals are typically understood as reflecting the aggregate activity of multiple hippocampal neurons, including both subthreshold and suprathreshold dynamics. Compared to the discrete nature of spike trains, which provide clearer event-like signals, LFP data often contain more complex and overlapping sources, making interpretation and decoding more difficult. While there are indeed documented cases where LFP signals yield higher decodability than spikes, these are often region- or task-specific. In our case, prior studies using the same dataset have shown clearer differentiation of correct and incorrect decisions using spiking data (Allen et al., 2016; Shahbaba et al., 2022). The goal of our work was to demonstrate that meaningful behavioral distinctions can also be extracted from LFP alone-a signal that is more challenging to work with in this context—highlighting the strength of our method.
> > >
> > > ## Regarding the spatial locations of the electrodes
> > > We appreciate the opportunity to clarify. Our method provides not only a measure of overall coherence between two brain regions, but also reveals the channel-wise contributions (Fig 5) to this coherence via the canonical vectors. This added layer of interpretability enables us to identify which specific channels are most responsible for driving inter-regional interactions, offering a more nuanced understanding of the underlying neural dynamics. We view this feature as a key innovation of the proposed framework, both in a general sense and also, as you rightly point out, in the particular context of our data application.
> > >
> > > ***We’re very glad*** to continue this discussion with you and greatly appreciate your engagement with our work.  We’d be happy to further clarify if there are any other aspects that remain unclear.

---

> > > > ### Comment · Reviewer_RXsG · 2025-08-06
> > > >
> > > > thank you. you’ve addressed all of my concerns and I’ve adjusted my score accordingly.

---

> > > > > ### Author Response · Authors · 2025-08-06
> > > > >
> > > > > We sincerely thank you for your thoughtful feedback and the time you dedicated to reviewing our work. We truly appreciate your positive evaluation, and we will carefully incorporate the insights from our discussion to further improve the paper.

---

### Note · Authors · 2025-08-12

We would like to take this opportunity to sincerely thank all reviewers and the Area Chair for your time, thoughtful feedback, and constructive suggestions throughout the review process. We are especially grateful for the recognition of the clarity and organization of our writing, the methodological rigor of both the theoretical development and experimental validation, and most importantly, the novelty of our contribution. In particular, we deeply appreciate the acknowledgment that WaveCanCoh is the first framework to extend Canonical Correlation Analysis (CCA) to the wavelet domain, enabling scale-specific inference on the dependence structure between multivariate nonstationary time series.

During the rebuttal phase, we carefully considered all reviewer comments and engaged in constructive, in-depth exchanges that we believe have clarified key aspects of our methodology and reinforced its significance. Based on these discussions, we are committed to making the following improvements in the revised version:

- Incorporating the Fourier-based (LSP) benchmark for real data to provide a clearer comparison and further highlight the advantages and novelty of WaveCanCoh.

- Expanding the discussion on limitations of the proposed method, including potential edge cases and directions for future methodological improvements.

- Enhancing the related work discussion to better differentiate WaveCanCoh from existing approaches and emphasize its contributions.

- Addressing some minor presentation issues, such as font size of several figures and notation consistency, to improve readability.

We are fully committed to contributing high-quality and impactful work to the community. As highlighted by several reviewers, WaveCanCoh presents a novel framework with strong potential for broad applications in neuroscience, deep learning techniques for time series. We believe this work deserves to be accepted to encourage further exploration and discussion within the community.

Once again, we sincerely thank all reviewers for your valuable comments and guidance, and the Area Chair for your support and coordination throughout the review process.

---

### Decision · Program_Chairs · 2025-09-17

**Decision:**

Accept (spotlight)

**Comment:**

This submission proposes a novel and theoretically sound method, Wavelet Canonical Coherence (WaveCanCoh), for analyzing dependencies in non-stationary multivariate time series, offering a significant advancement over existing techniques. While initial reviews raised valid questions regarding novelty and experimental breadth, the authors have convincingly addressed these concerns through detailed rebuttals and clarifications, resulting in significantly raised scores from multiple reviewers (a shift from borderline to positive recommendations). The work demonstrates a solid theoretical foundation, validated through both simulation and real-world data, and presents a promising approach for fields like neuroscience where analyzing dynamic signal relationships is crucial. Given the robust methodology, thoughtful responses to reviewer feedback, and potential impact, I recommend acceptance.